# Fibroblast-specific PRMT5 deficiency suppresses cardiac fibrosis and left ventricular dysfunction in male mice

Yasufumi Katanasaka [1,2,3] ✉, Harumi Yabe[1], Noriyuki Murata[1], Minori Sobukawa[1], Yuga Sugiyama[1], Hikaru Sato[1], Hiroki Honda[1], Yoichi Sunagawa[1,2,3], Masafumi Funamoto[1], Satoshi Shimizu[1], Kana Shimizu[1], Toshihide Hamabe-Horiike[1,2,3], Philip Hawke[4], Maki Komiyama[2], Kiyoshi Mori [3,5,6], Koji Hasegawa[1,2] & Tatsuya Morimoto [1,2,3] ✉

Protein arginine methyltransferase 5 (PRMT5) is a well-known epigenetic regulatory enzyme. However, the role of PRMT5-mediated arginine methylation in gene transcription related to cardiac fibrosis is unknown. Here we show that fibroblast-specific deletion of PRMT5 significantly reduces pressure overload-induced cardiac fibrosis and improves cardiac dysfunction in male mice. Both the PRMT5-selective inhibitor EPZ015666 and knockdown of PRMT5 suppress α-smooth muscle actin (α-SMA) expression induced by transforming growth factor-β (TGF-β) in cultured cardiac fibroblasts. TGF-β stimulation promotes the recruitment of the PRMT5/Smad3 complex to the promoter site of α-SMA. It also increases PRMT5-mediated H3R2 symmetric dimethylation, and this increase is inhibited by Smad3 knockdown. TGF-β stimulation increases H3K4 tri-methylation mediated by the WDR5/MLL1 methyltransferase complex, which recognizes H3R2 dimethylation. Finally, treatment with EPZ015666 significantly improves pressure overload-induced cardiac fibrosis and dysfunction. These findings suggest that PRMT5 regulates TGF-β/Smad3-dependent fibrotic gene transcription, possibly through histone methylation crosstalk, and plays a critical role in cardiac fibrosis and dysfunction.

Heart failure is a major global health problem, with an estimated 37.7 million patients worldwide[1]. This complex pathophysiological condition results from left ventricular remodeling, including cardiac hypertrophy and cardiac fibrosis. Cardiac fibrosis is the excessive deposition of extracellular matrix that develops in almost all chronic cardiac diseases, including myocardial infarction and hypertensive heart disease[2,3]. During the progression of cardiac fibrosis in heart failure, cardiac fibroblasts become activated and then differentiated into myofibroblasts, which are the major source of collagen fibers[4,5]. Activation of cardiac fibroblasts is mediated by inflammatory cytokines, neurohormone factors, and the stretching of the ventricular wall by myocardial infarction or pressure overload[6]. Thus, the inhibition of myofibroblast conversion is a promising therapeutic approach in fibrotic disease states including heart failure[7,8].

Changes in gene expression, central to the initiation of the fibrotic response, have been observed both in activated fibroblasts and in the

[1]Division of Molecular Medicine, School of Pharmaceutical Sciences, University of Shizuoka, Shizuoka, Japan. [2]Division of Translational Research, National Hospital Organization Kyoto Medical Center, Kyoto, Japan. [3]Shizuoka General Hospital, Shizuoka, Japan. [4]Laboratory of Scientific English, School of Pharmaceutical Sciences, University of Shizuoka, Shizuoka, Japan. [5]Graduate School of Public Health, Shizuoka Graduate University of Public Health, Shizuoka, Japan. [6]Department of Molecular and Clinical Pharmacology, School of Pharmaceutical Sciences, University of Shizuoka, Shizuoka, Japan. ✉e-mail: katana@u-shizuoka-ken.ac.jp; morimoto@u-shizuoka-ken.ac.jp

differentiation of quiescent fibroblasts into myofibroblasts in response to TGF-β, a key cytokine for the induction of fibrosis[9]. When TGF-β binds to its receptors, it induces the activation of canonical and non-canonical downstream signaling pathways. Khalil et al. reported that Smad3 is an essential transcription factor for cardiac fibroblast activation due to pathological cardiac stresses[10]. It has previously been suggested that TGF-β may be a direct target for anti-fibrosis therapy; however, the direct inhibition of TGF-β has been shown to lead to a variety of adverse effects[11]. Therefore, downstream mediation of the TGF-β signaling pathway is a potential alternative target for chronic heart failure therapy. A growing body of evidence suggests that epigenetic regulation of fibrotic gene expression by TGF-β/Smad3 signaling plays an important role in the development of heart failure, and that the key proteins regulated in the process have strong potential as a therapeutic target for heart failure[12].

The basic unit of chromatin is the nucleosome, which encompasses 147 base pairs of DNA wrapped around a histone octamer. Chemical modifications to nucleosomal DNA and their histone tails strongly influence gene transcription through epigenetic machinery[13]. Many previous studies have reported that epigenetic events, such as histone acetylation and methylation and DNA methylation, contribute to the pathogenesis of cardiac fibrosis in heart failure[12].

Arginine methylation is modified by the protein arginine methyltransferases (PRMTs), a family of 10 related molecules[14]. Type I, II, and III PRMTs catalyze asymmetric di-methylation (ADM), symmetric di-methylation (SDM), and mono-methylation of arginine, respectively[14]. A report by Yan F-Z et al revealed the involvement of PRMT1, a representative type I arginine methyltransferase, in liver fibrosis through the activation of hepatic stellate cells[15]. However, the function of PRMT5, a representative type II arginine methyltransferase, in tissue fibrosis has not yet been elucidated. PRMT5 is responsible for the majority of cellular SDM. It is known to methylate several targets, including histones (H4R3, H3R8, and H3R2) and transcriptional regulators (NF-κB and p53)[16,17]. Transcription by these regulators can be either driven or repressed, depending on which residues are modified within the tails of the histones. For example, while gene expression is repressed by the symmetric dimethylation of H4R3 and H3R8, it is promoted by the symmetric dimethylation of H3R2[18]. PRMT5 forms a protein complex with transcription factors, and its binding to promoter sites vary with the type of cell and stimulation. PRMT5 is essential for the proliferation and survival of cancer cells, and over-expression of PRMT5 is found in various types of human cancers[19]. In addition, it has been reported that PRMT5 functions in various other biological processes, including neuronal homeostasis, hematopoiesis, and metabolic syndrome. Chen H. et al. reported that the PRMT5/MEP50 methylosome complex regulates TGF-β-mediated epithelial-to-mesenchymal transition through histone arginine methylation[20]. In addition, the TGF-β/Smad3 signal pathway is essential for myofibroblast differentiation in tissue fibrosis[9]. However, the role of PRMT5 in cardiac fibrosis during the development of heart failure is still unknown.

The purpose of this study is to clarify the role of symmetric arginine methylation by PRMT5 both in the activation of cardiac fibroblasts and in the differentiation of these fibroblasts into myofibroblasts in vitro and in vivo. To investigate the functional role of the *Prmt5* gene in the development of heart failure, we generated fibroblast-specific PRMT5-KO mice. The results showed that a deficiency of *Prmt5* in fibroblasts prevented pressure overload-induced cardiac fibrosis and dysfunction in the mice. We also investigated the mechanism by which PRMT5 regulates TGF-β-stimulated cardiac fibroblast activation and differentiation into myofibroblasts, finding that PRMT5 interacted with Smad3 and regulated its dependent gene transcription through histone arginine dimethylation and lysine trimethylation.

## Results

### Pathological cardiac fibrosis is suppressed by PRMT5 depletion in activated fibroblasts

*Postn*[MCM] mice are frequently used in fibroblast-specific molecular analysis of pathological cardiac fibrosis, both because pressure overload induces pathological cardiac hypertrophy and fibrosis, and because periostin is preferentially expressed in activated cardiac fibroblasts[10,21,22]. In this study, to investigate the functional role of PRMT5 in pathological cardiac fibrosis, we examined pressure overload-induced cardiac fibrosis and dysfunction in *Postn*[MCM];*Prmt5*[flox/flox] mice (Fig. 1a). The mice were born in an expected Mendelian ratio (Supplementary Fig. 1). The PRMT5 floxed mice used in our experiment have previously been reported to reduce PRMT5 protein expression efficiently when crossed with CreERT mice after tamoxifen injection[17,23]. We examined the deletion of the PRMT5 gene with PCR, and western blotting showed that the expression of PRMT5 in fibroblasts was decreased (Fig. 1b, c). Echocardiography revealed both that TAC surgery led to a decrease in fractional shortening (FS) in *Prmt5*[flox/flox] mice, and also that this decrease was significantly lower in *Postn*[MCM];*Prmt5*[flox/flox] mice (Fig. 1d and Supplementary Table 1). While the baseline heart weight/body weight ratio was not changed by the knockout of *Prmt5* in fibroblasts, TAC-induced cardiac hypertrophy was suppressed in the *Postn*[MCM];*Prmt5*[flox/flox] mice (Fig. 1e, Supplementary Fig. 2). Histological analysis indicated that pressure overload-induced cardiac fibrosis was significantly decreased by *Prmt5* deletion in *Postn*-expressed fibroblasts (Fig. 1f). Additionally, mice expressing *Col1a2*[MCM], a fibroblast-specific gene[24], also exhibited the suppression of cardiac fibrosis induced by pressure overload (Fig. 1f, Supplementary Figs. 3 and 4, and Supplementary Table 2). Cardiac hypertrophy was not suppressed in the *Col1a2*[MCM];*Prmt5*[flox/flox] mice (Supplementary Fig. 5). Fibrotic gene expression was repressed in both *Postn*[MCM];*Prmt5*[flox/flox] and *Col1a2*[MCM];*Prmt5*[flox/flox] mice (Fig. 1h, i), supporting the data showing the inhibitory effect of fibrosis. We also examined cardiac vascularization, a known factor that affects myocyte contraction. The results showed that PRMT5 knockout in fibroblasts did not significantly change capillary density (Supplementary Fig. 6).

During the progression of cardiac fibrosis, fibroblasts differentiate into myofibroblasts and produce extracellular matrix proteins. Some reports have shown that myofibroblasts play a critical role in pathological cardiac fibrosis[12,25]. We examined the number of fibroblasts expressing α-SMA, a protein marker of myofibroblast differentiation, in the heart. The number of α-SMA positive fibroblasts was increased by TAC surgery in the *Prmt5*[flox/flox] mice, and *Prmt5* deletion in *Postn*-expressed fibroblasts significantly reduced this increase (Fig. 1j). These results suggest that, in activated cardiac fibroblasts, PRMT5 is required for pressure overload-induced pathological cardiac fibrosis and left ventricular dysfunction.

### PRMT5 is required for myofibroblast differentiation in cultured cardiac fibroblasts

As we had found that PRMT5 plays an important role in cardiac fibrosis and myofibroblast differentiation, we next examined the mechanism by which PRMT5 promotes myofibroblast differentiation. TGF-β is known to activate Smad3-related gene transcription and to initiate the differentiation of cardiac fibroblasts into myofibroblasts[10,26]. Pharmacological inhibition of PRMT5 with EPZ015666, a specific inhibitor of PRMT5 enzymatic activity[27], in cultured cardiac fibroblasts from adult humans and neonatal rats decreased the TGF-β-induced expression levels of *Col1a1* and *Acta2* mRNA, α-SMA protein, and proline incorporation (Fig. 2a, b, e, f, and Supplementary Fig. 7). Consistent with these results, siRNA knockdown of PRMT5 significantly reduced these expression levels (Fig. 2c, d, g, h). As the proliferation of fibroblasts is also characteristic of fibroblast activation, we also examined the proliferation of cardiac fibroblasts after PRMT5 knockdown and

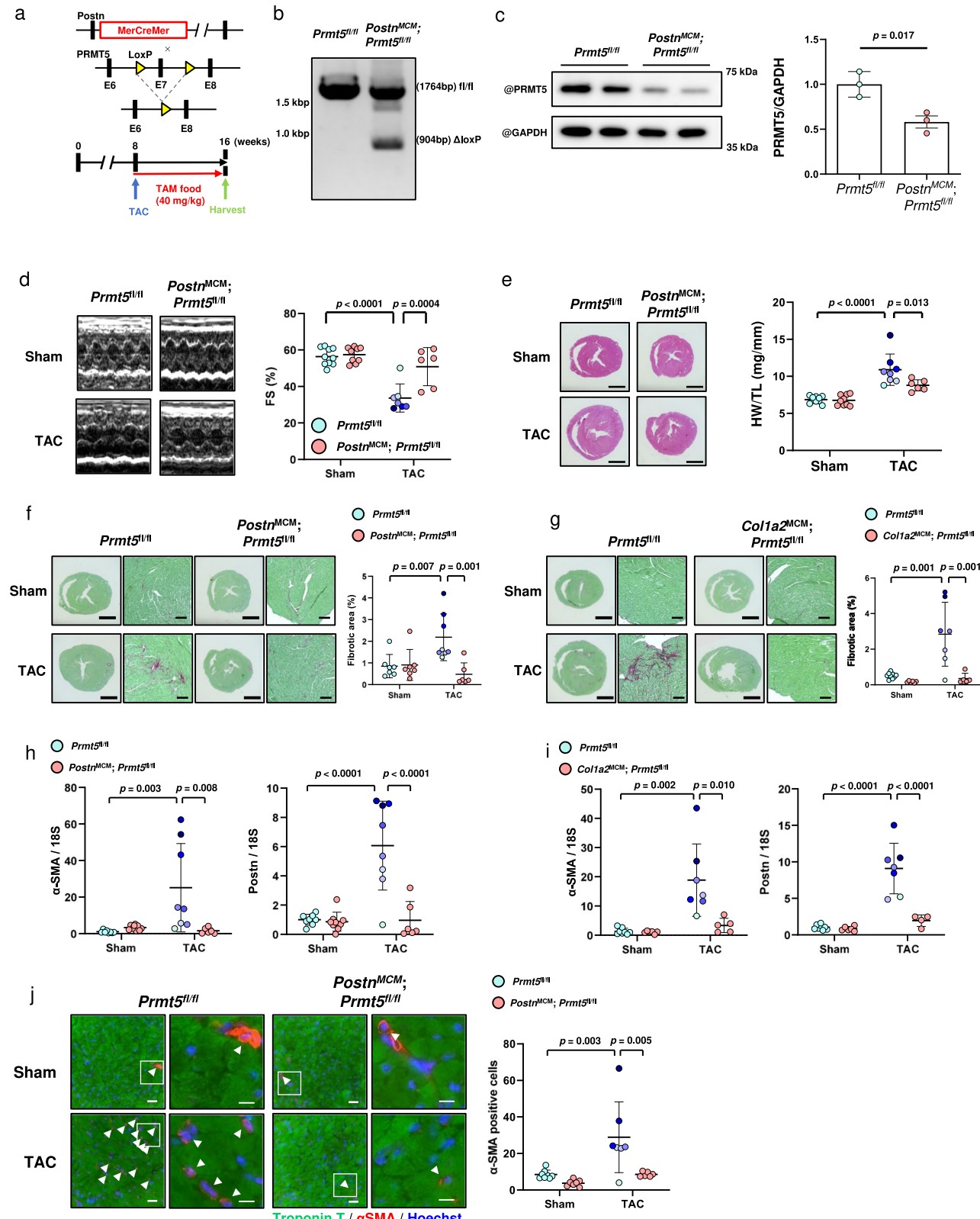

inhibition. The results showed that neither PRMT5 inhibition nor knockdown significantly affected fibroblast viability or proliferation in vitro (Supplementary Fig. 8). In addition, treatment with the PRMT5 inhibitor EPZ015666 did not significantly affect the viability or proliferation of adult cardiac fibroblasts in vitro (Supplementary Fig. 9).

PRMT5 regulates various biological processes through histone methylation, including H3R2 dimethylation, which is known to be a histone modification of transcriptional activation. We examined H3R2 dimethylation after TGF-β stimulation of cardiac fibroblasts. Treatment with EPZ015666 inhibited the activity of symmetric dimethylation in the fibroblasts, but not the total methylation of H3R2 (Supplementary Fig. 10). A ChIP assay revealed that symmetric H3R2 dimethylation at the *Col1a1* and *Acta2* promoter sites was significantly increased by TGF-β stimulation. These increases were suppressed by

**Fig. 1 | Prmt5 deficiency in fibroblast lineages attenuates pressure overload-induced cardiac fibrosis and dysfunction. a** Postn[MCM] lineage mice were crossed with Prmt5[fl/fl] mice to create fibroblast-specific Prmt5 knockout mice. After TAC surgery, chow containing tamoxifen (40 mg/kg/day) was administered to the mice for 8 weeks. **b, c** Knockout of Prmt5 in the fibroblasts of Postn[MCM];Prmt5[fl/fl] mice was confirmed by PCR and western blotting. Fibroblasts from tamoxifen-treated mice were cultured, and PRMT5 knockout was assessed by PCR and western blot analysis. **c** Values are presented as mean ± SD (n = 3 mice). **d** Echocardiographic analysis of Postn[MCM];Prmt5[fl/fl] and Prmt5[fl/fl] mice was performed at 8 weeks after TAC surgery, and fractional shorting was calculated. Values are presented as mean ± SD (n = 10 Prmt5[fl/fl] mice [sham], 9 Postn[MCM];Prmt5[fl/fl] mice [sham], 7 Prmt5[fl/fl] mice [TAC], and 6 Postn[MCM];Prmt5[fl/fl] mice [TAC]). **e** Prmt5 knockout in Postn-positive fibroblasts suppressed cardiac hypertrophy induced by pressure overload. Representative images of heart tissue are shown. Heart-weight-to-tibia-length ratio was calculated 8 weeks after TAC surgery. Values are presented as mean ± SD (n = 10, 9, 8, and 6 mice, respectively). **f, g** Representative images of hearts stained with picrosirius red (scale bars = 2 mm [whole], 100 μm [zoom]) in Postn[MCM];Prmt5[fl/fl] **f** and Col1a2[MCM];Prmt5[fl/fl] **g** mice at 8 weeks after TAC surgery. Fibrotic area (% heart area) was quantified with ImageJ software. Values are presented as mean ± SD (**f** n = 7, 7, 8, and 6 mice, respectively; **g** n = 9, 6, 7, and 5 mice, respectively). **h, i** Fibrotic gene expression (α-SMA and periostin) in hearts from Postn[MCM];Prmt5[fl/fl] **h** and Col1a2[MCM];Prmt5[fl/fl] **i** mice was quantified by qRT-PCR. Values are presented as mean ± SD (**h** n = 8, 9, 8, and 6 mice, respectively; **i** n = 8, 6, 7, and 5 mice, respectively). **j** Representative images of immunofluorescence co-stained with α-SMA (red), troponin T (green), and Hoechst 33342 (blue) (scale bars = 50 μm [whole], 10 μm [zoom]) for each group. Arrowhead indicates α-SMA⁺ fibroblasts. The number of α-SMA⁺ fibroblasts per field was counted with BZ-X Analyzer software (Ver. 1.1.2). Values are presented as mean ± SD (n = 8, 7, 7, and 6 mice, respectively). Two-way ANOVA, followed by Tukey's multiple comparison test. P values are indicated in each graph. Source data are provided as a Source Data file.

PRMT5 inhibition and knockdown (Fig. 2i, j). We also examined H4R3 dimethylation, which is a representative PRMT5 modification of histone and is known to repress gene transcription[19]. The results showed that H4R3 dimethylation was not significantly altered by TGF-β stimulation (Supplementary Fig. 11). These results suggest that PRMT5-mediated dimethylation of H3R2 may promote TGF-β-induced fibrotic gene transcription.

## PRMT5 interacts with Smad3

As it had been reported that Smad3 is required for TGF-β-induced cardiac fibrosis[10], we hypothesized that PRMT5 is associated with Smad3 and regulates the transcription of fibrotic genes such as Acta2 that are involved in myofibroblast differentiation. To test this hypothesis, we examined the interaction between PRMT5 and Smad3. Smad3 is a transcription factor that binds to a wide variety of protein molecules. Its activity is modulated by post-transcriptional modification and various binding partners such as transcriptional co-factors and co-repressors[28]. A GST pull-down assay revealed that His-tagged PRMT5 physically interacted with GST-tagged Smad3 (Fig. 3a). Moreover, PRMT5 bound to the Smad3 deletion mutant aa 225-425, which includes the MH2 domain that is essential for forming heterocomplexes with common-mediator Smads (Fig. 3b). IP-WB analysis showed that FLAG-tagged PRMT5 was associated with HA-tagged Smad3 in HEK293T cells (Fig. 3c, d). We also confirmed that PRMT5 interacted with Smad3 in TGF-β-stimulated cardiac fibroblasts using IP-WB analysis (Fig. 3e) and found that Smad3 was not methylated by PRMT5 in vitro (Supplementary Fig. 12). In addition, we examined PRMT5-Smad3 interaction with or without TGF-β stimulation in cardiac fibroblasts. This interaction was detected without TGF-β stimulation, and it did not change in the heart after TAC surgery (Supplementary Fig. 13), suggesting that TGF-β stimulation was not a trigger of the interaction. We evaluated the expression of PRMT5 in mouse heart after TAC surgery. The results showed no significant change in PRMT5 expression (Supplementary Fig. 14a). Additionally, PRMT5 expression in phenylephrine-treated cardiomyocytes and TGF-β-treated cardiac fibroblasts was not significantly altered (Supplementary Figs. 14b, c). These results suggest that PRMT5 expression does not change in cardiomyocytes or cardiac fibroblasts under heart failure conditions.

After undergoing TGF-β stimulation, Smad3 is known to translocate from the cytoplasm to the nucleus, where it binds to genomic DNA[29]. To determine whether PRMT5 is also recruited to Smad binding sites on genomic DNA, we performed a ChIP analysis of TGF-β-treated cardiac fibroblasts. We found that PRMT5 was indeed recruited to the Smad binding sites, and that this enhanced recruitment was significantly inhibited by the knockdown of Smad3 (Fig. 3f). In addition to PRMT5 recruitment, H3R2 dimethylation at Col1a1 and Acta2 promoter sites induced by TGF-β stimulation was significantly inhibited by Smad3 knockdown (Fig. 3g). These results suggest that PRMT5 is recruited onto Smad-binding elements by Smad3 and methylates histones.

We also examined the phosphorylation and reporter activity of Smad3, finding that the inhibition of PRMT5 did not affect TGF-β induced phosphorylation of Smad3 or its dependent promoter activity (Supplementary Fig. 15). Additionally, we examined the phosphorylation level of SMAD3 in mouse heart and found that there was no difference between the WT and PRMT5 KO groups (Supplementary Fig. 16). p38 MAPK signaling is another factor known to play an important role in fibroblast activation[21]. However, in our study, the use of a PRMT5 inhibitor did not markedly suppress the phosphorylation of p38 in cardiac fibroblasts (Supplementary Fig. 17). Taken together, these findings support the view that PRMT5-mediated histone dimethylation is required for fibrotic gene transcription induced by the TGF-β/Smad3 pathway.

## WDR5/MLL1-mediated H3K4 tri-methylation is required for myofibroblast differentiation

As the above results suggested that H3R2 dimethylation was induced by TGF-β stimulation in cardiac fibroblasts, we next considered the role of H3K4 trimethylation in fibrotic gene transcription. H3R2 symmetric dimethylation has been reported to induce H3K4 trimethylation, which is known to activate gene transcription, through the WDR5/MLL1 lysine methyltransferase complex[30–32]. We therefore hypothesized that H3R2 dimethylation induces H3K4 trimethylation in TGF-β-activated cardiac fibroblasts. With a ChIP assay, we examined the characteristics of H3K4 trimethylation at the Smad binding site in the region that codes fibrotic genes. In cardiac fibroblasts, TGF-β treatment increased H3K4 trimethylation during myofibroblast differentiation. This increase in histone trimethylation was significantly inhibited by treatment with MM102, a WDR5/MLL1 inhibitor (Fig. 4a). Importantly, this trimethylation was also significantly reduced by treatment with a PRMT5 inhibitor (Fig. 4b). To determine the effect of PRMT5 on MLL1 methyltransferase activity, we examined total H3K4 trimethylation. The results showed that PRMT5 inhibition did not significantly change the total amount of H3K4 trimethylation in cardiac fibroblasts (Supplementary Fig. 18). These results suggest that TGF-β-induced H3K4 trimethylation may be mediated by PRMT5-induced H3R2 dimethylation, which is recognized by the WDR5/MLL1 complex in cardiac fibroblasts.

We then investigated the role of the WDR5/MLL1 complex in myofibroblast differentiation. Pharmacological inhibition of WDR5−MLL1 interaction using MM102 significantly suppressed TGF-β-induced increases both in the mRNA levels of Col1a1 and Acta2 and in the protein expression of α-SMA in cardiac fibroblasts (Fig. 4c, d). Similar results were also observed in experiments using siRNA for MLL1 and WDR5 (Fig. 4e, f, and Supplementary Fig. 19). These results suggest that the WDR5/MLL1 methyltransferase complex is required for myofibroblast differentiation in cardiac fibroblasts.

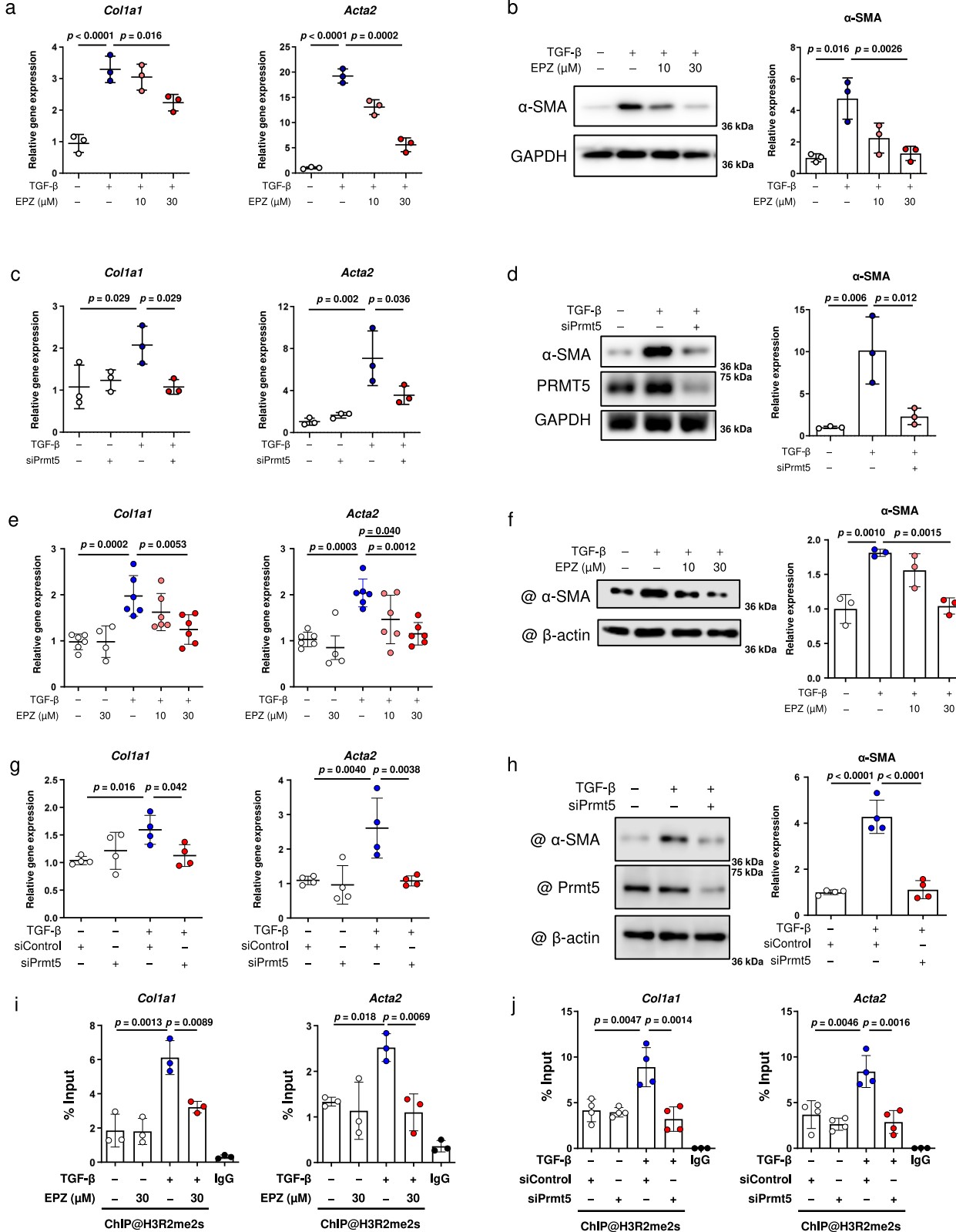

## Pharmacological inhibition of PRMT5 suppresses pathological cardiac fibrosis

The therapeutic use of PRMT5-specific inhibitors is under development, and a variety of ongoing cancer clinical trials are investigating their pharmacokinetics as well as their beneficial and adverse effects. As we found that *Prmt5* deficiency in fibroblasts suppressed cardiac dysfunction, we hypothesized that PRMT5 inhibitors have anti-heart

failure effects via their inhibition of cardiac fibrosis. We examined the pharmacological effect of a PRMT5 inhibitor on a mouse model of pressure overload-induced heart failure. Previous research has shown that the PRMT5 inhibitor EPZ015666 is applicable to in vivo studies without significant adverse effects[27,33,34]. EPZ015666 (30 mg/kg/day) and vehicle were orally administered for 8 weeks to mice that had undergone TAC or sham surgery (Fig. 5a). There was no significant

**Fig. 2 | Pharmacological inhibition or knockdown of PRMT5 suppresses TGF-β-induced fibrotic gene expression. a–d** Primary cultured cardiac fibroblasts from adult human ventricle were treated with EPZ015666 at the indicated concentration **a**, **b** or siRNA knockdown of PRMT5 **c**, **d** and then treated with or without TGF-β (10 ng/mL). **e–h** Primary cultured cardiac fibroblasts from neonatal rats were treated with EPZ015666 at the indicated concentration **e**, **f** or siRNA knockdown of PRMT5 **g**, **h** and then treated with or without TGF-β (10 ng/mL). Gene expression levels of *Col1a1* and *Acta2* were quantified by qRT-PCR **a**, **c**, **e**, **g**. The protein expression level of α-SMA was determined by western blotting and quantified with

ImageJ software **b**, **d**, **f**, **h**. **i**, **h** Chromatin immunoprecipitation (ChIP) of histone H3 symmetric dimethylarginine and qPCR analysis of the precipitated genomic DNA at the promoter regions of *Col1a1* and *Acta2* were performed with EPZ015666-treated **i** and siRNA-transfected **j** cardiac fibroblasts. Immunoprecipitation with IgG was used as a negative control for ChIP. Values are presented as mean ± SD. **a–d**, **i** $n = 3$ biologically independent samples; **e** $n = 6$ biologically independent samples; **g**, **h** $n = 4$ biologically independent samples. One-way ANOVA, followed by Dunnett's multiple comparison test versus TGF-β treated group **a**, **b**, **e**, **f** or Tukey's multiple comparison test **c**, **d**, **g**, **h–j**. Source data are provided as a Source Data file.

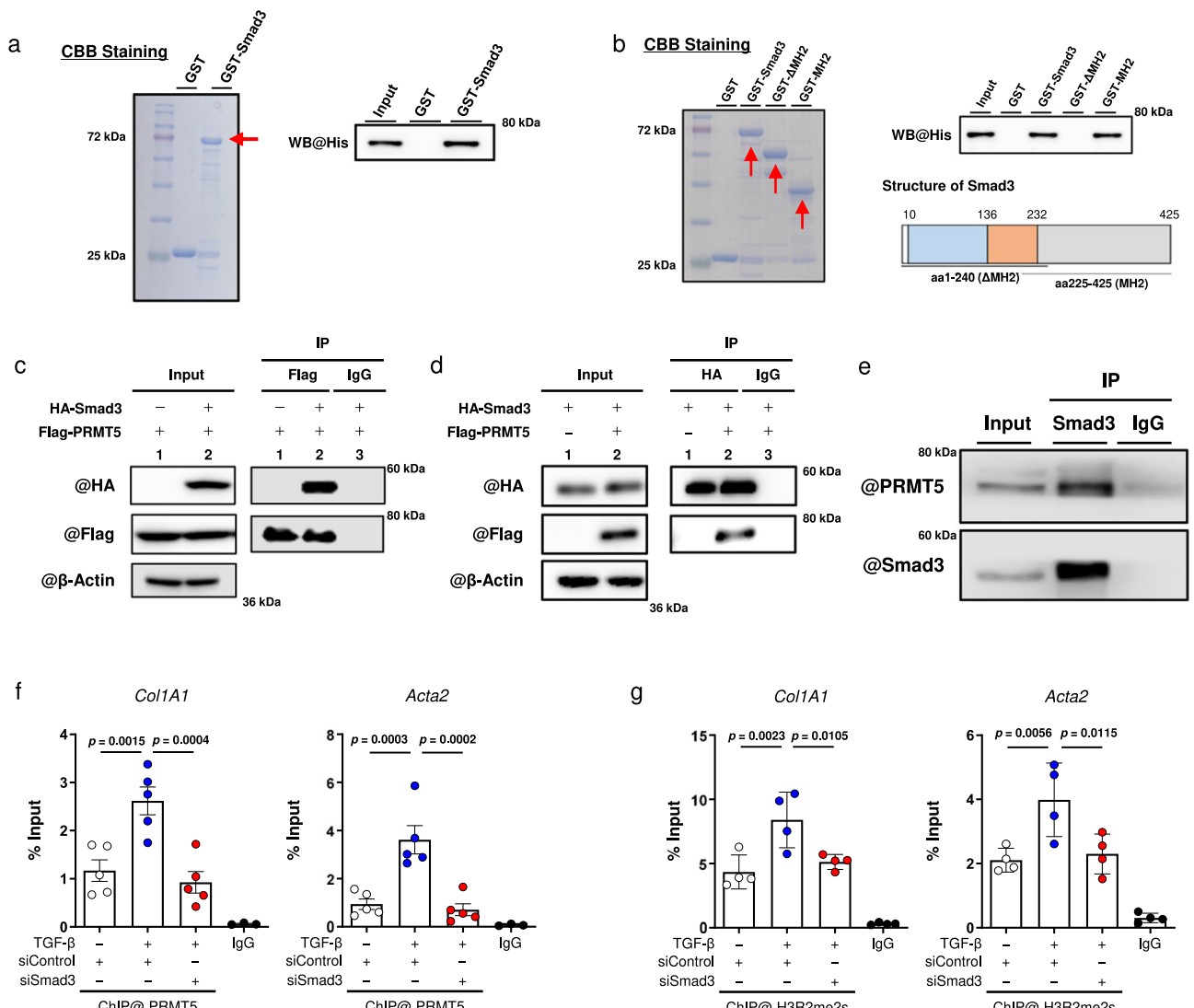

**Fig. 3 | PRMT5 binds directly to Smad3 in cardiac fibroblasts. a** The interaction between recombinant full-length GST-Smad3 and his-tagged PRMT5 was examined by GST pull-down assay. Results of CBB staining and western blotting are shown. Arrows show GST-fusion proteins. **b** The interaction between recombinant full-length or deletion mutants of GST-Smad3 and his-tagged PRMT5 was examined by GST pull-down assay. Results of CBB staining and western blotting are shown. Arrows show GST-fusion proteins. Schematic diagram of Smad3 domains is included. **c**, **d** Flag-tagged PRMT5 and HA-tagged SMAD3 were transfected into HEK293T cells. Immunoprecipitation with anti-Flag **c** or anti-HA **d** beads was performed, then western blotting was performed with indicated antibodies. **e** Primary

cultured cardiac fibroblasts from neonatal rats were stimulated with TGF-β (10 ng/mL) for 2 h. Smad3 was immunoprecipitated and western blotting was performed with indicated antibodies. **a–e** Similar results were obtained from three biologically independent samples. **f**, **g** Chromatin immunoprecipitation and quantitative PCR analysis were performed with anti-PRMT5 antibody or anti-histone H3R2me2 antibody to determine their degree of recruitment to the *Col1a1* and *Acta2* promoter sites. IgG was used as a negative control. Values are presented as mean ± SD. **f** $n = 5$ biologically independent samples; **g** $n = 4$ biologically independent samples. One-way ANOVA, followed by Tukey's multiple comparison test **f**, **g**. *P* values are indicated in each graph. Source data are provided as a Source Data file.

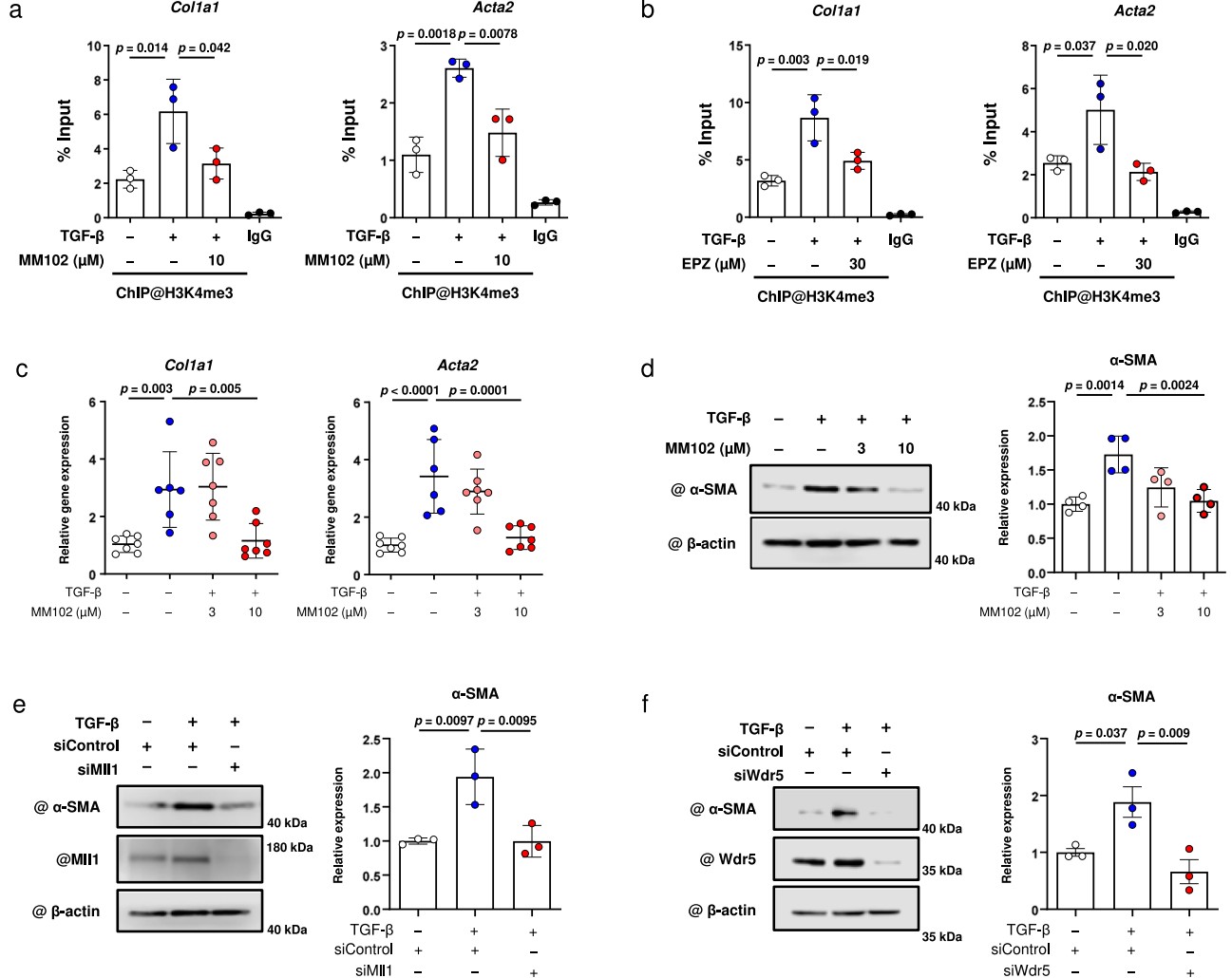

**Fig. 4 | EPZ015666 inhibits lysine tri-methylation of histone, and WDR5/MLL1 lysine methyltransferase is required for TGF-β-induced fibrotic gene regulation. a, b** TGF-β induces WDR5/MLL1-mediated H3K4 trimethylation, which is inhibited by PRMT5. Primary cultured cardiac fibroblasts from neonatal rats were treated with or without TGF-β (10 ng/mL) in the presence or absence of MM102 **a** or EPZ015666 **b** for 6 h. Chromatin immunoprecipitation with anti-histone H3K4me3 antibody was performed, followed by quantitative PCR analysis of the *Col1a1* and *Acta2* promoter sites. IgG was used as a negative control. **c–f** Primary cultured cardiac fibroblasts were treated with MM102 at the indicated concentration and then treated with or without TGF-β (10 ng/mL). Gene expression levels of *Col1a1*

and *Acta2* were quantified by qRT-PCR **c**. The protein expression level of α-SMA was determined by western blotting and quantified with ImageJ software **d**. Primary cultured cardiac fibroblasts were transfected with siRNA (siControl, siMLL1, or siWDR5) and then treated with or without TGF-β (10 ng/mL). The protein expression of α-SMA was quantified by western blotting **e–f**. Values are presented as mean ± SD. **a, b, e, f** $n = 3$ biologically independent samples; **c** $n = 6$ biologically independent samples; **d** $n = 4$ biologically independent samples. One-way ANOVA, followed by Tukey's multiple comparison test (a, b, e, f) or Dunnett's multiple comparison test versus TGF-β-treated group **c, d**. *P* values are indicated in each graph. Source data are provided as a Source Data file.

change in body weight after treatment with EPZ015666 (Supplementary Fig. 20). Echocardiographic analysis showed that the decrease in fractional shortening (FS) and the increase in left ventricular posterior wall thickness (LVPWd) induced by TAC surgery were significantly improved by treatment with EPZ015666 (Fig. 5b). Cardiac hypertrophy was significantly decreased in the EPZ015666-treated group compared with the vehicle group (Fig. 5c). Histological analysis showed that EPZ015666 treatment suppressed increases in cardiomyocyte cell hypertrophy and interstitial fibrosis (Fig. 5c, d). The expression of fibrosis-related genes (*Acta2*, *Col1a1*, and *Postn*) was significantly increased by pressure overload, but this upregulation was significantly repressed by treatment with EPZ015666 (Fig. 5e). A previous study reported that conditional knockout of PRMT5 in cardiomyocytes caused dilated cardiomyopathy through the dysregulation of O-GlcNAcase (OGA)[35]. We therefore examined the cell viability of cultured cardiomyocytes from neonatal rats after EPZ015666 treatment. The results did not show a significant decrease in cell viability or OGA

expression (Supplementary Figs. 21 and 22). These results indicate that pharmacological inhibition of PRMT5 suppresses pressure overload-induced pathological cardiac hypertrophy, fibrosis, and left ventricular dysfunction.

## Discussion

TGF-β/Smad3 signal-mediated fibrotic gene transcription plays a central role in myofibroblast differentiation and cardiac fibrosis. Smad3 cooperates with various proteins, including epigenetic regulators, to regulate gene transcription[28]. DNA methylation and histone lysine acetylation and methylation are involved in the process of myofibroblast differentiation from cardiac fibroblasts[12]. However, prior to the present study, the role of arginine methylation in cardiac fibroblasts had not been investigated. This study demonstrates (1) that knockout of *Prmt5* in fibroblasts suppresses both cardiac fibrosis and dysfunction induced by pressure overload, (2) that PRMT5 interacts with Smad3, and that this interaction is required for the recruitment of

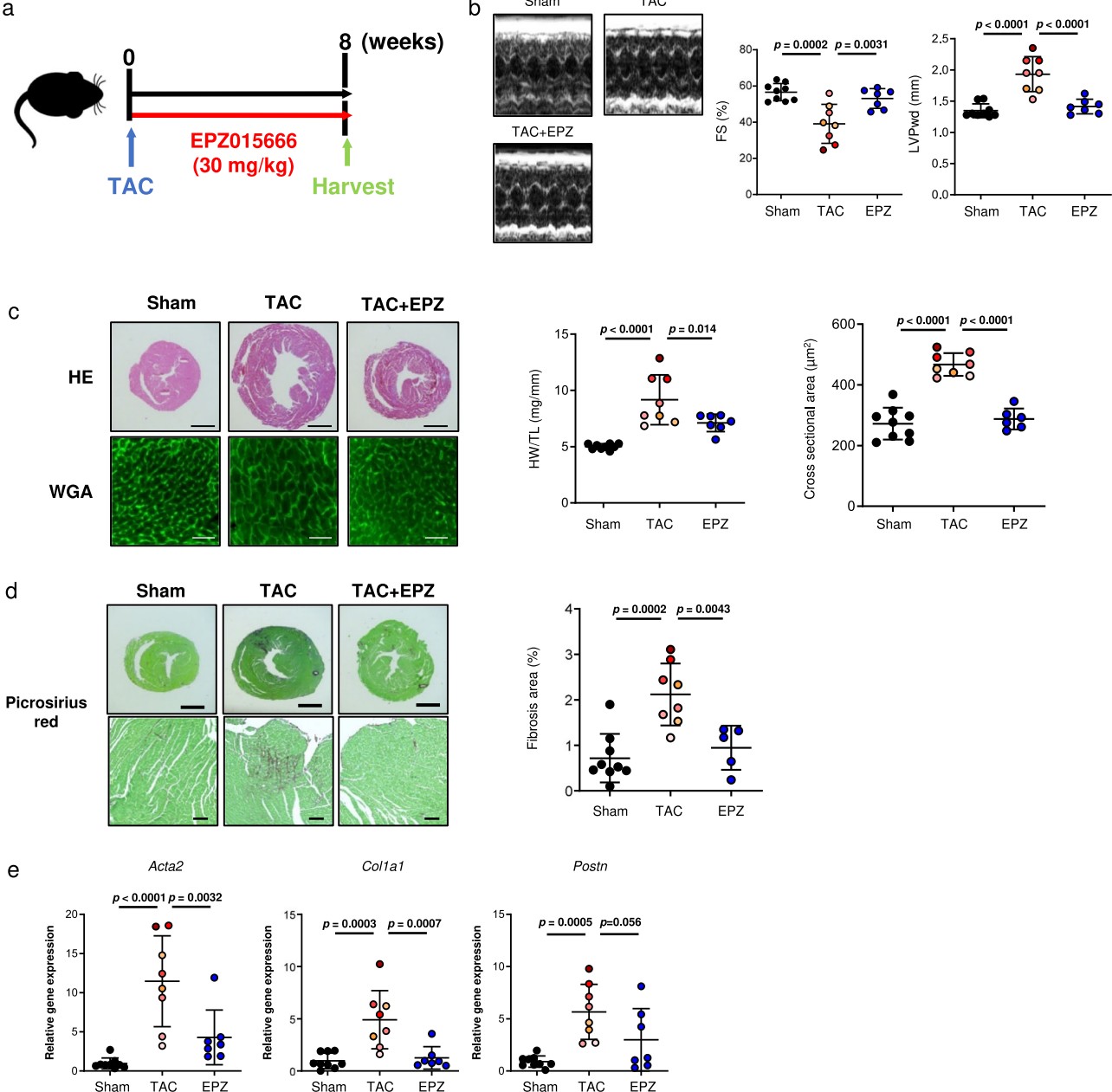

**Fig. 5 | EPZ015666 suppresses pressure overload-induced cardiac fibrosis and dysfunction. a** C57BL/6j male mice (8-10 weeks old) were subjected to TAC or sham surgery and then orally administered with EPZ015666 (30 mg/kg/day) or vehicle for 8 weeks. **b** Echocardiographic analysis was performed at 8 weeks after TAC surgery, and fractional shorting (FS) and left ventricular posterior wall thickness at diastole (LVPWd) were calculated. **c** Representative images of HE and WGA staining (scale bars = 2 mm [whole], 50 μm [zoom]) in each group. The ratio of heart weight to tibial length (HW/TL) was calculated. The cross-sectional area of cardiomyocytes was quantified with BZ-X Analyzer software (Ver. 1.1.2). **d** Representative images of picrosirius red staining (scale bars = 2 mm [whole], 100 μm [zoom]) in each group. The percentage total fibrosis area of the mouse heart after surgery was quantified with ImageJ software. Values are presented as mean ± SD. Sham, *n* = 9 mice; TAC, *n* = 8; EPZ015666 treatment, *n* = 5. **e** Heart mRNA was isolated, and fibrotic gene expression of *Acta2*, *Col1a1*, and *Postn* were quantified by qRT-PCR. Values are presented as mean ± SD. Number of mice in **b**, **c**, and **e**: Sham, *n* = 9; TAC, *n* = 8; EPZ015666 treatment, *n* = 7. One-way ANOVA, followed by Dunnett's multiple comparison test. *P* values are indicated in each graph. Source data are provided as a Source Data file.

PRMT5 onto the promoter sites of fibrotic genes, (3) that both PRMT5-mediated histone arginine methylation and WDR5/MLL1-mediated histone lysine methylation play important roles in fibrotic gene transcription in cardiac fibroblasts, and (4) that pharmacological inhibition of PRMT5 ameliorates cardiac fibrosis and dysfunction. These results suggest that PRMT5 plays a crucial role in pressure overload-induced cardiac fibrosis, possibly through the initiation of crosstalk between histone arginine methylation and lysine methylation (Fig. 6).

It is well known that the manipulation of genes found specifically in target cells is useful for analyzing the tissue-specific molecular functions of that gene under physiological and pathophysiological conditions. The type of cell that disease-associated cardiac fibroblasts develop from has been much discussed in recent papers. Two research groups have reported that periostin-expressing fibroblasts in the heart play an important role in pathological cardiac fibrosis in heart failure. The ablation of periostin-expressing cardiac fibroblasts with genetically modified mice prevents the induction of cardiac fibrosis and dysfunction by myocardial infarction and angiotensin II[36,37], suggesting that the inhibition of critical signal pathways in these fibroblasts ameliorates heart failure. Khalil et al. have demonstrated that the

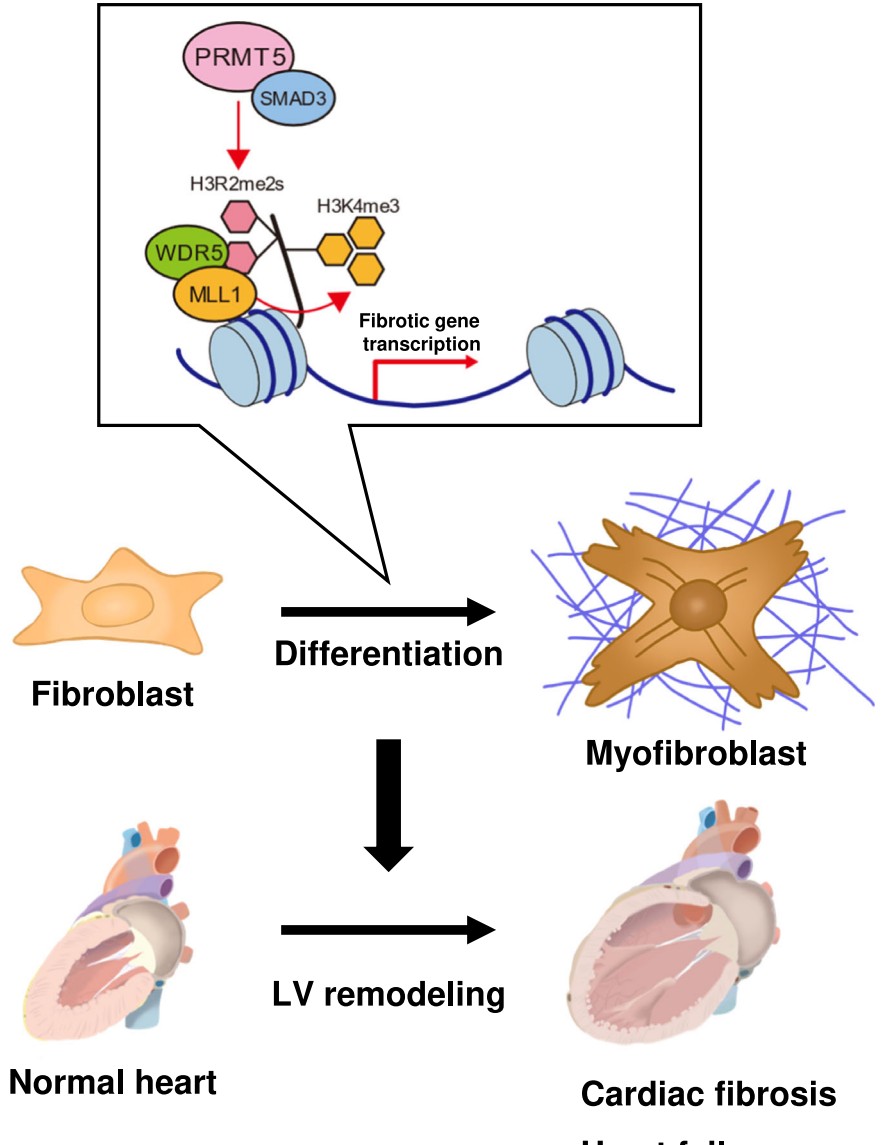

**Fig. 6 | Schematic illustration.** In the promoter regions of fibrotic genes, PRMT5 interacts with Smad3 and symmetrically dimethylates histone H3R2, and WDR5/MLL1 subsequently induces H3K4 trimethylation. These histone methylations trigger fibrotic gene transcription in cardiac fibroblasts, contributing to the progression of heart failure.

TGF-β/Smad3-mediated signal pathway in periostin-expressing fibro-blasts is required for pressure overload-induced cardiac fibrosis[10]. Moreover, it has been reported that both p38MAPK and β-catenin signaling are required for the activation of periostin-expressing fibroblasts[10,21,22]. These previous findings suggest that the functional proteins in periostin-expressing fibroblasts are a potential molecular target for a pharmacological therapy for heart failure. Our results show that *Prmt5* knockout in both *Postn*- and *Col1a2*-positive cells resulted in reduced fibrosis after pressure overload, indicating the importance of PRMT5 in the development of cardiac fibrosis.

Our results also show that cardiac hypertrophy is suppressed by PRMT5 deficiency specifically in periostin-positive cells. The knockout of TGF-β receptors 1/2 and β-catenin in periostin-expressing fibroblasts significantly reduced cardiac hypertrophy induced by pressure overload[10,22]. Moreover, the interaction between cardiomyocytes and fibroblasts has been proposed in the progression of heart failure[38]. In line with these previous studies, our results indicate that PRMT5 in periostin-expressing fibroblasts plays a key role in the development of cardiac hypertrophy. As cardiac fibrosis is regulated by various factors, including fibroblast proliferation, further mechanistic studies are needed to analyze the fibrosis-related functions of PRMT5.

TGF-β is the primary cytokine driving fibrosis, not only in the heart but also in various other tissues such as those in the kidney, liver, and lungs[30,31]. Smad3 is involved in the TGF-β-dependent canonical signaling pathway inducing cardiac fibrosis[31]. It is well known that Smad3 is a fundamental transcription factor inducing collagen and α-SMA expression. Smad3 interacts with multiple proteins and regulates gene transcription in various cells under many conditions[28]. Post-transcriptional modification of Smad3 alters both its binding to other proteins and its transcriptional activity[28]. Many transcriptional coactivators and corepressors, including epigenetic writers and erasers, are known to interact with Smad3 and to regulate the activation and repression of its transcriptional activity through histone modification[11]. In this study, we have found that PRMT5, which participates in the symmetric di-methylation of arginine residues in proteins, binds to Smad3. Moreover, we have found that this interaction is

essential for both PRMT5 recruitment and histone arginine methylation at the Smad3 target region of genomic DNA.

Previous studies have shown that histone methylation controls other modifications that regulate gene transcription[27]. PRMT5-mediated dimethylation of H3R2 has been reported to be a marker for the recruitment of WDR5/MLL1, which catalyzes the tri-methylation of H3K4 in cancer cell lines[27,32]. Our findings provide evidence for the importance of cooperation between the histone H3R2 di-methylation induced by PRMT5 and the H3K4 trimethylation induced by WDR5/MLL1 in the regulation of TGF-β/Smad3-mediated gene transcription in cardiac fibroblasts. This finding naturally raises the question of whether PRMT5 also directly methylates and thereby activates Smad3, but we have shown that this is not the case, as the in vitro methylation assay carried out in this study revealed that Smad3 was not methylated by PRMT5. In addition, to our knowledge there are no previous reports of Smad3-mediated transcriptional activity being regulated by methylation. In summary, Smad3 recruits PRMT5 onto its target region, and epigenetic modification induced by PRMT5 plays a key role in the transcriptional activation that occurs during fibroblast activation and myofibroblast differentiation.

A variety of PRMT5 inhibitors have been developed and are currently being examined in clinical trials for cancer therapy[33]. The PRMT5 selective inhibitor EPZ015666 is a first-in-class compound with a selectivity for PRMT5 that is over 20,000 times higher than the selectivity of the other PRMTs[25]. This inhibitor has shown an anti-tumor effect in mantle cell lymphoma in experimental mouse models, without severe adverse effects. Our study also showed that treatment with EPZ015666 for 8 weeks did not have any severe adverse effects. More importantly, two previous reports have demonstrated that treatment with PRMT5 inhibitors did not cause unmanageable side effects in patients with solid tumors in Phase I clinical trials[34,35]. Our results showed that the pharmacological inhibition of PRMT5 suppressed both cardiac hypertrophy and fibrosis in vivo. A previous study also reported on the function of PRMT5 in cardiomyocytes, finding that conditional knockout of PRMT5 in cardiomyocytes caused dilated cardiomyopathy through dysregulation of protein O-GlcNAcylation, suggesting that PRMT5 maintains cardiac homeostasis[36]. Although further studies, especially in the form of clinical trials, are needed to clarify the safety and optimize the administration schedule of PRMT5 inhibitors, the present study indicates that therapeutic strategies targeting PRMT5 are of considerable interest for chronic heart failure therapy.

Previous research has also revealed that WDR5/MLL1 is a methyltransferase complex, and that the somatic mutation of MLL1 is involved in acute and chronic leukemia[39]. Chemical inhibitors that disrupt the protein-protein interaction of WDR5 and MLL1 have been developed and have been shown to be potential anti-tumor agents for leukemia[40]. To our knowledge, the function of the WDR5/MLL1 methyltransferase complex in cardiac fibrosis, and in the TGF-β/Smad signal pathway in particular, remains unknown. It has been reported that pharmacological inhibition of WDR5/MLL1 reduces renal senescence in fibroblasts induced by ischemia reperfusion[37]. Together with the results of the present study, these findings suggest that WDR5/MLL1 may also be a target for the amelioration of cardiac fibrosis and heart failure.

In conclusion, we have shown that the epigenetic enzyme PRMT5 in cardiac fibroblasts plays a critical role in the progression of pressure overload-induced cardiac fibrosis and dysfunction. In the fibroblasts, PRMT5 binds to Smad3 and modifies histone arginine methylation at Smad target sites. After that, the lysine methyltransferase WDR5/MLL1 is recruited to the PRMT5-dependent arginine methylation site, where it trimethylates H3K4. PRMT5-induced arginine methylation and MLL1-induced lysine methylation may then initiate transcription of the α-SMA gene,

inducing myofibroblast differentiation. These findings elucidate the regulatory mechanisms of gene transcription associated with myofibroblast differentiation and may contribute to the development of a pharmacotherapy targeting epigenetic enzymes such as histone arginine and lysine methyltransferases.

## Methods

### Materials
Human PRMT5 plasmid vector was kindly gifted by Dr. Naoya Fujita[41], and pCMV Flag hSmad3 vector was purchased from Riken BRC (RDB07019, Tsukuba, Japan). The PCR product of the PRMT5 sequence was cloned into a pET28a vector (Merck, Tokyo, Japan) with an N-terminal His-tag and an entry vector[42]. Then the sequence was sub-cloned into pcDNA3.2/V5-DEST using Gateway Technology (Thermo Fisher Scientific, Tokyo, Japan). The PCR product of Smad3 mutants (aa1-240 and aa225-425) was cloned into a pGEX6P1 vector (GE Healthcare, Tokyo, Japan) using the restriction enzymes BamHI and XhoI (Thermo Fisher Scientific). EPZ015666 was purchased from DC Chemicals (Shanghai, China). MM102 was purchased from Selleck Chemicals (S7265, Houston, TX, USA). Tamoxifen was purchased from APExBIO (B5965, Houston, TX, USA). The following small interfering RNAs were purchased from Sigma-Aldrich (Tokyo, Japan): Mission siRNA Universal Negative Control, Mission siRNA for rat PRMT5 (SASI_Rn02_00242865), rat WDR5 (SASI_Rn02_00204788), and rat MLL1 (SASI_Rn02_00239409).

### Mice
The study protocol complied with the guidelines on animal experiments approved by the Institutional Animal Care and Use Committee at the University of Shizuoka (#196404) and the National Hospital Organization Kyoto Medical Center (#2-25-3). The protocol includes justification of the use of animals, their welfare, and the incorporation of the principles of the "3Rs" (Replacement, Reduction, and Refinement). All animals were maintained in a pathogen-free facility at room temperature 23 ± 1 °C in 12:12 h light and dark cycles. The animals were housed in microisolator cages on individually ventilated cage racks filled with aspen chip bedding. The animals were euthanized by cervical dislocation at the end of each animal experiment. Male C57BL/6j mice (8 weeks old) were purchased from Japan SLC (Shizuoka, Japan). C57BL/6 *Col1A2*[Mer-Cre-Mer (MCM)] mice (Stock # 029567) and *Periostin* (*Postn*)[MCM] mice (Stock #029645) were obtained from Jackson Laboratories (Bar Harbor, ME, USA). PRMT5-floxed mice (EMMA ID: 07883) were obtained from Infrafrontier Research Infra-structure (München, Germany). To generate a fibroblast-specific PRMT5-KO mouse model, *Col1A2*[MCM] mice (*Col1A2*[MCM];*PRMT5*[fl/fl]) and *Periostin*[MCM] mice (*Postn*[MCM];*PRMT5*[fl/fl]) were bred with PRMT5-floxed mice. In all experiments, male *PRMT5*[fl/fl] mice were used as a control for the PRMT5-KO mice. All of the mice were male and were fed a standard diet (CE-2). *Postn*[MCM] mice received continuous tamoxifen-citrate chow feeding (40 mg/kg body weight; TD.130860, Envigo, Tokyo, Japan–) until the experiment was terminated[10]. *Col1A2*[MCM] mice were injected intraperitoneally with 40 mg/kg tamoxifen for 10 consecutive days[24]. All of the mice in our experiments were treated with tamoxifen, according to the same protocol. The number of mice included in each study is indicated in the figures or their legends.

### Transverse aortic constriction (TAC) models
Eight- to ten-week-old male mice were anesthetized with 2% isoflurane, and their limbs were fixed. Under artificial respiration (0.3 mL, respiratory rate of 150 per min), the chest was incised to the second intercostal space. Then, under a microscope, the aortic arch was doubly ligated with a 27-gauge needle using 6-0 silk thread, and the intercostal space and skin were sutured with 6−0 nylon thread[43]. In sham surgery, the same procedure was performed except for the ligation.

## Echocardiography

Cardiac function was assessed using a 10–12 MHz probe (S12, Philips, Amsterdam, Netherlands) and ultrasonic diagnostic equipment (Envision-C, Philips) at 8 weeks after TAC surgery[44]. Mice were anesthetized with isoflurane at a concentration of 3-5% (induction phase) and 1% (maintenance phase). Left ventricular diameter at diastole (LVIDd, mm), left ventricular diameter at systole (LVIDs mm), and left ventricular posterior wall thickness at diastole (LVPWd, cm) were measured with M-mode images. Fractional shortening (FS, %) was calculated as [LVIDd - LVIDs] / LVIDd × 100 (%). LV mass was calculated as $1.055 [(IVSd + LVIDd + LVPWT)^3 - (LVIDd)^3]$. LV mass index is represented as the ratio of LV mass to body weight. Measurements were performed by three individuals who were blinded to the identity of the experimental groups of the mice using Image J software.

## Histological analysis

Heart tissues were harvested and sliced across the papillary muscles, then fixed with 10% formalin and embedded in paraffin. Five-μm-thick sections were de-paraffinized and rehydrated. After rehydration in 70% ethanol, the sections were incubated in filtered Mayer's hematoxylin solution (Cat# 131-09665, Wako, Osaka, Japan) and washed with tap water. Then the sections were incubated in eosin solution (Cat# 051-06515, Wako), dehydrated with an ethanol series (70%, 80%, 90%, and 100%), cleared in xylene, and mounted in MGK-S mounting medium (Cat# FK00500, Matsunami, Osaka, Japan). For picrosirius red staining, the sections were incubated with freshly prepared staining buffer (1.2% picric acid [Cat #88-89-1, Wako], 0.1% Fast Green FCF [Cat #F7252, Sigma-Aldrich], and 0.1% Direct Red 80 [Cat #365548, Sigma-Aldrich]) for 1 h at room temperature. Sections were washed briefly in distilled $H_2O$ and dehydrated. The slides were mounted in MGK-S mounting medium. Images were taken with a Leica M165C microscope (Wetzlar, Germany) to examine global change in heart size. The images were analyzed with Image J software to quantify fibrosis. Interstitial fibrosis was calculated by removing the perivascular fibrosis area is expressed as a percentage of the total tissue section area[43].

## Cardiomyocyte cross-sectional area (CSA) analysis

Heart sections were stained with Alexa Fluor 488 conjugated-WGA (Cat #C10607, Thermo Fisher, Tokyo, Japan). An operator who was blinded to mouse genotype quantified cardiomyocyte CSA using computer-assisted morphometric analysis of microscopy images acquired on a BZ-X710 microscope (Keyence, Tokyo, Japan) and analyzed using BZ-X Analyzer software version 1.1.2 (Keyence)[43]. The average CSA of 50-60 randomly selected round-shaped cardiomyocytes per section was used for analysis.

## Immunofluorescence staining

Antigen activation was performed by boiling in ethylenediaminetetraacetic acid (EDTA)-based solution. The sections were blocked with 3% bovine serum albumin for 30 min and then incubated with anti-rabbit troponin T antibody (Cat #26592-1-AP, Proteintech, Tokyo, Japan) at 1:100 dilution for 1.5 h at room temperature to stain the cardiomyocytes. After the slides were washed, the sections were incubated in Alexa647 goat anti-rabbit IgG (Cat #A-21244, Thermo Fisher Scientific) at 1:500 dilution for 30 minutes at room temperature. Sections were then incubated with mouse monoclonal anti-α-SMA antibody (Cat #A5228, Sigma Aldrich) at 1:500 dilution and Alexa555 goat anti-mouse IgG (Cat #A-21429, Thermo Fisher Scientific) at 1:500 dilution. Isolectin GS-IB4, Alexa Fluor™ 647 Conjugate (Cat#I32450, Thermo Fisher Scientific) was used to stain vasculature. Hoechst 33258 solution (Cat #23491-45-4, Wako) was used to stain the nucleus. The sections were mounted with Fluoromount-G mounting medium (Cat #00-4958-02, Thermo Fisher Scientific). The number of α-SMA positive cells was counted in five high-magnification fields (x400) per tissue. α-SMA positive cells around blood vessels were eliminated from the analysis.

## qRT-PCR

TRI Reagent (Sigma-Aldrich, Cat #TR118) was added to the cells and homogenized tissues. Chloroform (Cat #67-66-3, Wako) was added and centrifuged (12,000 x g, 4 °C, 15 mins). 2-Propanol (Cat #67-63-0, Wako) was added to the supernatant, and the mixture was centrifuged (12,000 x g, 4 °C, 10 min). RNA pellets were washed with 75% ethanol (Wako). After air drying, the pellets were dissolved with DNase/RNase-free water. The extracted mRNA was reverse transcribed using ReverTra Ace qPCR RT Master Mix (Cat #FSQ-201, Toyobo, Osaka, Japan) according to the manufacturer's instructions. Quantitative RT-PCR was performed using KOD SYBR qPCR Mix (Toyobo) according to the manufacturer's instructions using a LightCycler 96 Real-Time PCR System (Roche, Tokyo, Japan). The primers shown in Supplementary Table 3 were used for the analysis. After the reaction, a threshold value was set at an appropriate position on the obtained amplification curve, and the number of cycles at the intersection with the amplification curve was taken as the cycle threshold (Ct) value. The amount of PCR product was obtained from the difference in Ct value from the control group, normalized to the 18 S value[45].

## Cell culture

HEK293T cell lines (CRL-3216, American Tissue Culture Collection, Maryland, USA) were maintained in Dulbecco's minimal essential medium (DMEM, Nacalai Tesque, Kyoto, Japan) supplemented with streptomycin (100 μg/mL), penicillin (100 units/mL), and 10% heat-inactivated fetal bovine serum (FBS) at 37 °C under 5% $CO_2$ in a humidified chamber. Adult human cardiac fibroblasts were purchased from PromoCell (Cat#C-12375, Heidelberg, Germany) and cultured according to the manufacturer's instructions. To prepare primary cultured cardiac fibroblasts, hearts from 1- to 3-day-old SD rats were removed, minced, and incubated with collagenase (Worthington Industries, Worthington, OH, USA) / pancreatin (Sigma-Aldrich) in Hank's Balanced Salt Solution (HBSS, Nacalai Tesque) at 37 °C. After incubation for 1 h, the medium was removed, and the attached cells were further cultured in 10% FBS DMEM. Cardiac fibroblasts were pretreated with EPZ015666 for 2 h in serum-free DMEM and then stimulated with transforming growth factor-β1 (TGF-β1, PeproTech, Cranbury, NJ, USA) at a final concentration of 10 ng/mL. Small interfering RNA was transfected with Lipofectamine RNAiMAX (Thermo Fisher Scientific) according to the manufacturer's instructions with some modifications[46].

## GST pull-down assay

Recombinant proteins were prepared from BL21DE3 competent *E. coli* cells. GST fusion proteins were immobilized on glutathione-Sepharose 4B beads (GE Healthcare) and mixed with a His-tag fusion human PRMT5 protein. The mixtures were gently rotated at 4 °C for 2 h. Then the beads were washed four times, and the binding proteins were eluted with SDS-PAGE sample buffer (125 mM Tris-HCl, pH 6.8, 4% sodium SDS, 10% 2-mercaptoethanol, 20% glycerol, and 0.002% bromophenol blue) and separated by SDS-PAGE. GST fusion proteins were visualized with Coomassie brilliant blue staining, and the binding proteins were detected by western blotting with anti-His-tag mAb (MBL Life Science, Tokyo, Japan)[42].

## Immunoprecipitation

Nuclear extracts were prepared from HEK293T cells and primary cultured cardiac fibroblasts. For immunoprecipitation, anti-FLAG M2 agarose affinity gel (Cat #A2220, Sigma-Aldrich) or anti-Smad3 rabbit monoclonal antibody (Abcam, Tokyo, Japan) were used. Normal rabbit IgG was used as a negative control (Jackson Immuno-Research Laboratories, West Grove, PA, USA). The appropriate antibodies were added to the nuclear extracts, and the mixtures were gently rotated at 4 °C for 2 h. The beads were washed four times, and the immunocomplexes were eluted with SDS sample buffer.

## Western blotting

The extracted proteins were subjected to SDS-PAGE and then transferred to nitrocellulose membrane (Hybond-ECL, GE Healthcare). After blocking with 5% skim milk in PBS-T (0.05% Tween20 in PBS), the proteins were reacted with primary and secondary antibodies[46,47]. The band images were taken with an Amersham Imager 680 (Cytiva, Tokyo, Japan). Quantification was performed using ImageJ software. The following primary antibodies were used: anti-PRMT5 rabbit monoclonal antibody (Cat#07-405, Merck, Tokyo, Japan) at 1:5000 dilution, anti-Smad3 rabbit monoclonal antibody (#9523, Cell Signaling Technology, Tokyo, Japan) at 1:2000 dilution, anti-WDR5 rabbit monoclonal antibody (#13105, Cell Signaling Technology) at 1:2000 dilution, anti-MLL1 rabbit monoclonal antibody (#14197, Cell Signaling Technology) at 1:2000 dilution, anti-α-SMA mouse monoclonal antibody (Cat#A5228, Sigma-Aldrich) at 1:5000 dilution, anti-HA-tag rabbit monoclonal antibody (Cat#M132-3, MBL Life Science) at 1:10000 dilution, anti-FLAG-tag rabbit monoclonal antibody (Cat#M185-3L, MBL Life Science) at 1:10000 dilution, anti-MGEA5 polyclonal antibody (Cat#14711-1-AP, Proteintech) at 1:5000 dilution, Anti-Histone H3 (trimethyl K4) rabbit monoclonal antibody (Cat#ab8580, Abcam) at 1:5000 dilution, and anti-β-actin mouse monoclonal clone AC-15 IgG (Cat#A1978, Sigma-Aldrich) at 1:10000 dilution. The following secondary antibodies were used: goat anti-rabbit IgG-HRP (Cat#458, MBL Life Science) at 1:10000 dilution. and goat anti-mouse IgG-HRP (Cat#330, MBL Life Science) at 1:10000 dilution.

## L-proline cell uptake measurement

After TGF-β stimulation of cultured cardiac fibroblasts, L-proline [$^3$H] (0.25 µCi, Moravek Biochemicals, Brea, CA, USA) was added to the cells, and then the cells were incubated at 37 °C for 48 h. Next, the cells were washed 3 times with PBS and precipitated with 5% trichloroacetic acid (TCA, Wako). After incubation at room temperature, they were rewashed with TCA and solubilized with 0.5 N NaOH (Wako), and the lysis solution was neutralized with an equal amount of 0.5 N HCl (Wako). Radioactivity was measured with a scintillation counter (LSC-7400, Aloka, Tokyo, Japan).

## Chromatin immunoprecipitation (ChIP)

Primary cultured cardiac fibroblasts were treated with or without TGF-β for 2 or 6 h. After fixation of genomic DNA and nuclear proteins with formalin, the cellular extract was sonicated, and the DNA-protein complex was immunoprecipitated with antibodies as follows: anti-PRMT5 rabbit monoclonal antibody (Active Motif, Tokyo, Japan), anti-Smad3 rabbit monoclonal antibody (Abcam), anti-H3R2me2s rabbit monoclonal antibody (Merck), or goat IgG[48]. The immunocomplexes were collected by incubation with protein A or G beads. After the precipitates were washed four times, the DNA was eluted and purified by phenol-chloroform extraction and ethanol precipitation. Quantitative PCR analysis was performed with KOD SYBR qPCR Mix (Toyobo) using a LightCycler 96 Real-Time PCR System. PCR was performed using the primers shown in Supplementary Table 3.

## Statistical analysis

Data are expressed as mean ± SD. The Shapiro-Wilk normality test was used to evaluate data distribution. Statistical analyses were evaluated by parametric analysis: unpaired (two-tailed) Student's t-test (with Welch's correction when variance was unequal) for two groups, and one-way ANOVA with Dunnett's or Tukey's multiple comparison test for three groups or more. Data with more than one variable were evaluated by two-way ANOVA with post-hoc Tukey's multiple comparison tests. All statistical analysis was performed with GraphPad Prism 9 software (GraphPad Software). The data generated in this study are provided in the Source Data file.

## Reporting summary

Further information on research design is available in the Nature Portfolio Reporting Summary linked to this article.

## Data availability

All data supporting the findings of this study are available within the paper and its Supplementary Information. Source data are provided in this paper.

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

## Acknowledgements

This work was supported by a JSPS Grant-in-Aid for Scientific Research (C) Grant Number JP19K07325 (Y.K.). It was also supported in part by grants from the MSD Life Science Foundation, the Public Interest Incorporated Foundation (19-570, Y.K.), the Takeda Science Foundation (17-552, Y.K.), the SENSHIN Medical Research Foundation (21-561, Y.K.), and the Mochida Memorial Foundation for Medical and Pharmaceutical Research (20-562, Y.K.).

## Author contributions

Katanasaka Y, Yabe H, Murata N, Sugiyama Y, Sobukawa M, Sato H, Honda H, Funamoto M, Shimizu S, Shimizu K and Sunagawa Y conducted experiments and acquired the data. Hamabe-Horiike T and Komiyama M analyzed the data. Katanasaka Y, Hawke P, and Morimoto T wrote the manuscript. Mori K and Hasegawa K provided supervision. N. Katanasaka Y and Morimoto T provided funding.

## Competing interests

The authors declare no competing interests.
