## [Peer Review File · Nature Communications]

REVIEWER COMMENTS

Reviewer #1 (Remarks to the Author):

This study investigated the role of PRMT5 in cardiac fibrosis and found that fibroblast-specific deletion of PRMT5 significantly reduced pressure overload-induced cardiac fibrosis and improved cardiac dysfunction. Mechanically, PRMT5 regulate TGF- β /Smad3-dependent fibrotic gene transcription through histone methylation. This investigation that should be of interest to the field. However, some of the important experiments are lacking to support the main idea, several issues should be addressed to strengthen the working model. These are several comments outline below will hopefully be useful for the authors to consider.

1. It is not clear why the authors choose the study the role of PRMT5 in cardiac fibrosis? Why choose to study PRMT5 but other genes? Does the expression of PRMT5 changed in cardiomyocyte and in cardiac fibroblast in response to TAC surgery?

2. Fig 2, without TGF β stimulation, does PRMT5 inhibition and knockdown modulate the H3R2 dimethylation at the fibrosis-associated genes' promoter sites? PRMT5 is responsible for the methylation of many histone proteins like histone H2A, H3, H4, and as well as non-histone proteins, why only H3R2 dimethylation examined? Some of them might also be important for the fibroblast activation and regulated by PRMT5, therefore contribute to the observed protective effects of PRMT5 on cardiac fibrosis.

3. To characterize the activation of fibroblast, CF proliferation should be examined. In addition to differentiation, is PRMT5 required for CF proliferation?

4. It is interesting that the cardiac hypertrophy was suppressed in the Postn MCM/Prmt5 flox/flox mice but not in the col1a2 MCM/Prmt5 flox/flox mice. The authors should explain. In addition, does PRMT5 regulate the methylation level or expression of Col1a2?

5. Fig 3c-3d, the interaction between SMAD3 and PRMT5 has been verified by IP-WB in HEK293T at basal level. I wonder whether this direct interaction can still exist in cardiac fibroblasts without TGF β stimulation. If not, why? If so, what is the upstream regulatory mechanism that recruited PRMT5 to SMAD3 binding site under TGF β stimulation. Speculatively, if the interaction is specifically happened under TGF β stimulation, Figure S8, does PRMT5 methylate Smad3 in vivo under TGF β stimulation?

6. Figure S9C, it seems that the phosphorylation of SMAD2/3 bands in TGF β +EPZ015666 upregulated? The statistical data in right should be carefully checked. Thus, it should be caution with the interpretation of the currently proposed model. More much evidence should be provided. e.g. the phosphorylation level of SMAD3 in Prmt5 knockdown mice with and without TAC surgery.

7. The authors claims that PRMT5-mediated H3R2 dimethylation is required for myofibroblast differentiation in cardiac fibroblasts. However, limited data directly support this point. The data currently only support that PRMT4 can affect the deposition of H3R2 dimethylation on Col1a1 and Acta2, the

expression of Col1a1 and Acta2 are regulated by intact PRMT5. However, whether the expression change of Col1a1 and Acta2 are directly regulated by PRMT5-mediated H3R2 dimethylation is unknown.

8. PRMT5 mediated H3R2me2s modification may activate transcription via recruitment of WDR5/MLL and the subsequent H3K4me3 modification. It may also function via other pathway. Fig. 4, even the H3K4 trimethylation deposition was affected by PRMT5 inhibitor, we cannot get the conclude that H3K4 trimethylation is mediated by PRMT5-induced H3R2 dimethylation. To support the proposed conclusion in this section, more experiment data should be provided. It would be better to provide data to confirm (1) the H3K4 trimethylation change in fibroblast is directly regulated by H3R2 dimethylation on those examined genes, like Cola1, Acta2. (2) the contribution of TGF β stimulation should be clarified, whether H3K4 trimethylation deposition which is induced by H3R2 dimethylation are only happened in myofibroblast differentiation (or TGF β -stimulation dependent) or not?

Minor:

1. Fig 1d, it seems that Col1a2 MCM/Prmt5 flox/flox mice decreased fibrosis at Sham group. Is it statistically significant?

Reviewer #2 (Remarks to the Author):

In this paper entitled "Fibroblast-specific PRMT5 deficiency suppresses pressure overload-induced cardiac fibrosis and left ventricular dysfunction", authors declared that PRMT5 regulates TGF- β /Smad3-dependent fibrotic gene transcription through histone methylation crosstalk and plays a critical role in cardiac fibrosis and dysfunction. However, the data are not elucidating the contribution of fibroblast for the observed phenotypes and several results are hard to interpret with the given information. Specific comments can be found below.

Major concern:

1. PostnMCM/Prmt5flox/flox and Col1a2MCM /Prmt5 flox/flox mice were performed in Figure 1; however, it is required for the baseline level of mouse organs, including the heart, and the fluorescence and expression evidence of fibroblast specific knockout.
 2. Cre mice have cardiotoxicity, so a group of cre mice need to be added as the control.
 3. Cardiac Echocardiography data display is insufficient, such as left ventricular mass index.
 4. The authors only evaluated the level of fibrosis in Figure 1, ignoring myocardial hypertrophy. Whether PostnMCM/Prmt5flox/flox and Col1a2MCM /Prmt5 flox/flox mice subjected TAC displayed myocardial hypertrophy compared with flox mice ?
- 4.a-SMA is the marker of myofibroblasts. It is also expressed in mural cells, and more markers are needed as fibroblast marker, refer to the article (Cell Rep

. 2018 Jan 16;22(3):600-610. doi: 10.1016/j.celrep.2017.12.072.)

5. The role of PRMT5 in phenotypic transformation of fibroblasts was showed in Figure2; actually, the fibrotic phenotype is not only phenotypic transformation, but also includes proliferation and collagen contraction.
6. In the pressure overload-induced cardiac fibrosis model, it is wrong for the author to focus only on the transformation from fibroblasts to myofibroblasts. In the article (Circulation. 2020 Oct 13;142(15):1448-1463. doi: 10.1161/CIRCULATIONAHA.119.045115. Epub 2020 Jul 30.), activated fibroblast emerged after induction of tissue stress to promote fibrosis instead of smooth muscle actin-expressing myofibroblasts, a key profibrotic cell population.
7. How did the authors determine smad3 as the target of prmt5? In fact, the fibrosis pathway also includes p38, yap, etc.
8. H3R2 symmetric dimethylation has been reported to induce H3K4 trimethylation; The trimethylation induced fibrosis gene transcription was inhibited by a PRMT5 inhibitor and WDR5/MLL1 inhibitor. These results cannot prove the regulation of PRMT5/WDR5/MLL1 methylase complex. The author first needs to prove that prmt5 regulates WDR5/MLL1 methylase comple, and demethylase also needs to be excluded.
9. EPZ015666 suppresses pressure overload-induced cardiac fibrosis and dysfunction. EPZ is a prmt5 inhibitor, acting on several kind of cells including fibroblasts. The therapeutic strategy of fibroblast specific targeting PRMT5 needs to be applied.

Reviewer #3 (Remarks to the Author):

In this manuscript the authors nicely demonstrated deletion of Prmt5 in cardiac fibroblasts reduced cardiac fibrosis after TAC-induced hypertrophic remodeling. Furthermore, they have showed that PRMT5 interacted with Smd3 and promoted α SMA expression in response to Tgfb using fibroblast cell culture model. Finally, they showed that pharmacological inhibition of PRMT5 has similar cardiac phenotype as Prmt5 KO.

Major concerns:

- 1). Hypertrophic remodeling is a hallmark of TAC injury. The authors showed that fibroblast-deletion of Prmt5 reduced TAC-induced cardiac fibrosis. How about post-TAC hypertrophic remodeling? e.g, HW/BW, cardiomyocyte size, hypertrophic gene expression (ANF, BNP, α / β MHC), etc. ? Are they reduced by Prmt5-deletion as well?
- 2) Like the authors pointed out in the discussion, PRMT5's function in cardiomyocytes is opposite from that of fibroblast; deletion of Prmt5 in cardiomyocytes caused dilated cardiomyopathy. What about pharmacological inhibition of PRMT5 using PMRT5 inhibitor EPZ015666? Since EPZ015666 is administrated systematically, it should inhibit PRMT5 in cardiomyocytes as well? In figure 5 experiments,

do you see changes in OGA expression and cardiac protein O-GlcNacylation caused by EPZ-015666 treatment?

Minor concerns.

1) Figure 1g, aSMC staining in sham hearts appears to be from smooth muscle cells around vessel rather than from myofibroblasts. It may be difficult to identify aSMA-expressing myofibroblasts under baseline condition since myofibroblasts are not activated. This may need to be clarified.

2) The mechanistic studies in figure 2-4 were done with primary cardiac fibroblasts. For Nat communication, I think to confirm some of the results in the hearts may be necessary.

Reviewer #4 (Remarks to the Author):

The manuscript addresses the interesting and highly relevant therapeutic concept of preventing cardiac remodeling and heart failure by preventing cardiac hypertrophy and fibrosis. The authors demonstrate that a fibroblast specific deletion of PRMT5 protects from adverse left ventricular remodeling after pressure overload induced by TAC in mice. The mechanism is claimed to be that PRMT5 forms a complex with SMAD3 after TGF beta stimulation, which increases symmetric dimethylation of target gene promoters. This gets in turn recognized by WDR5/MLL1 which regulates TGF-beta induced fibrotic gene regulation. Finally, they switch back to an in vivo model and show a beneficial effect of PRMT5 inhibition by EPZ015666 after TAC.

While this is an important topic the authors need to place more emphasis on clearly demonstrating that their main claim, which is the induction of fibrosis and hypertrophy by PRMT5, is true. This would require substantial additional experimental evidence in order to underscore the author's hypothesis.

There are several major points which need to be addresses:

1. The authors have to show that the PRMT5 knockout is working! If this is the initial description and use of this floxed mouse line, they need to show the reduction in PRMT5 protein and that no truncated protein is expressed that could be partially functional. If the line has been published previously, the relevant publication showing knockout efficiency must be cited.

2. How efficient is the tamoxifen-induced knockout with the POSTN and the Col1a2 MerCreMer lines in the authors' hands after TAC surgery? What's the percentage of myofibroblasts in which the PRMT5 gene was deleted? This is important information to assess the role of PRMT5 in preventing heart remodeling

after pressure overload. Also, did the controls receive tamoxifen as well (either in their chow or by injection)?

3. A major concern is the huge variability of the PRMT5^{fl/fl} control group after TAC! This is true for all experiments, and this reviewer wonders if the surgery was not performed consistently. What was the survival rate after TAC with a 27G needle? The degree of interstitial fibrosis is very low for a TAC with a 27G needle. Why is this so? Also FS should be less than 30% after 8 weeks. It would be helpful to correlate individual data points (colour coded), so that it is possible to see the values of the outliers in Figs 1 and 5! Is the large variability in the data maybe due to a difference between the sexes? The authors need to indicate whether they used exclusively male or female mice. In Fig. 5 it is mentioned that 8-10 weeks old male C57BL/6j mice were used. What about Fig. 1 where the largest variability takes place?

4. for the in-vitro experiments primary cardiac fibroblasts were taken from 1 to 3 days old rats. Why did the authors change the model system from mouse to rat? Why were fetal cardiac fibroblasts used? Are they a good model for adult cardiac fibrosis? Postnatal fibroblasts are different from adult fibros in terms of expression, proliferative capacity etc. To draw appropriate conclusions, the authors need to verify the data obtained with neonatal rat fibroblasts with adult cardiac fibroblasts. This is an important concern because all mechanistic conclusions were drawn from in vitro experiments.

5. The number of aSMA-positive fibroblasts was increased after TAC surgery in the controls but to a lesser degree in the Postn-specific Prmt5-knockouts. What is the cause of this phenomenon? Is fibroblast proliferation affected or is apoptosis increased?

the authors need to stain for markers of proliferation and apoptosis and to quantify their results. It is possible that PRMT5 induces fibroblast proliferation, as has been shown in retinoblastoma, and that its main action is via cell cycle genes rather than genes involved in fibrosis. This is very likely because fewer myofibroblasts mean less fibrosis.

6. Is there an effect on capillary density in the PRMT5 knockouts? It has been shown that a better vascularization can prevent a maladaptive response.

7. There is an issue with some of the statistics. In Fig. 2b,d; 4d,e,f and Suppl. Fig.9B the data had been normalized to control samples, which had been given arbitrary values of 1 with zero variance. This data is unsuitable for a one-way ANOVA. The authors need to normalize correctly.

Minor points:

1. Complete immunoblot images are missing. Band size markers are missing for all immunoblots.
2. Fig. 4e Labeling of the blots does not match labeling of the quantification.
3. EMMI: 07883 should be EMMA.
4. I found a preprint of the manuscript deposited and publicly available in "Research Square", so that a double blind review was not possible.

Responses to Reviewers

Ref.: Manuscript ID NCOMMS-22-45978

Fibroblast-specific PRMT5 deficiency suppresses pressure overload-induced cardiac fibrosis and left ventricular dysfunction

Reviewer 1

Reviewer 1: General comment

This study investigated the role of PRMT5 in cardiac fibrosis and found that fibroblast-specific deletion of PRMT5 significantly reduced pressure overload-induced cardiac fibrosis and improved cardiac dysfunction. Mechanically, PRMT5 regulate TGF- β /Smad3-dependent fibrotic gene transcription through histone methylation. This investigation that should be of interest to the field. However, some of the important experiments are lacking to support the main idea, several issues should be addressed to strengthen the working model. These are several comments outline below will hopefully be useful for the authors to consider.

Response: We are grateful to Reviewer 1 for the critical comments and useful suggestions that have helped us improve our paper. As indicated in the following responses, we have incorporated all of these comments and suggestions in the revised version of our manuscript.

Comment 1

It is not clear why the authors choose the study the role of PRMT5 in cardiac fibrosis? Why choose to study PRMT5 but other genes? Does the expression of PRMT5 changed in cardiomyocyte and in cardiac fibroblast in response to TAC surgery?

Response: We appreciate the reviewer's comment. PRMT5 is a type II enzyme that catalyzes arginine methylation and epigenetically regulates the transcription of various genes. It has been reported that epigenetic regulation is essential for cardiac fibrosis, especially myofibroblast differentiation¹. With this background, we hypothesized that PRMT5 plays a role in cardiac fibroblasts and chose to study the function of PRMT5 in cardiac fibrosis. We have revised the Introduction section to explain our hypothesis more clearly as follows:

"A report by Yan F-Z et al revealed the involvement of PRMT1, a representative type I arginine methyltransferase, in liver fibrosis through the activation of hepatic stellate cells.

However, the function of PRMT5, a representative type II arginine methyltransferase, in tissue fibrosis has not been elucidated." (Page 4, Lines 58-61)

We evaluated the expression of PRMT5 in mouse heart after TAC surgery. The results did not show a significant change in PRMT5 expression. Additionally, PRMT5 expression in hypertrophied cardiomyocytes (induced by phenylephrine) and cardiac myofibroblasts (induced by TGF- β) was not significantly altered in vitro. These results suggest that PRMT5 expression does not change in cardiomyocytes or cardiac fibroblasts under heart failure conditions. We have added these results to Supplemental Figure 14 of the revised manuscript.

"We evaluated the expression of PRMT5 in mouse heart after TAC surgery. The results showed no significant change in PRMT5 expression (Supplementary Figure 14a). Additionally, PRMT5 expression in phenylephrine-treated cardiomyocytes and TGF- β -treated cardiac fibroblasts was not significantly altered (Supplementary Figures 14b and c). These results suggest that PRMT5 expression does not change in cardiomyocytes or cardiac fibroblasts under heart failure conditions." (Pages 8-9, Lines 168-174)

1. Felisbino MB, McKinsey TA. Epigenetics in Cardiac Fibrosis: Emphasis on Inflammation and Fibroblast Activation. *JACC Basic Transl Sci.* 2018;3:704-715.
2. Yan F-Z, Qian H, Liu F, Ding C-H, Liu S-Q, Xiao M-C, Chen S-J, Zhang X, Luo C, Xie W-F. Inhibition of protein arginine methyltransferase 1 alleviates liver fibrosis by attenuating the activation of hepatic stellate cells in mice. *The FASEB Journal.* 2022;36:e22489

Comment 2

Fig 2, without TGF β stimulation, does PRMT5 inhibition and knockdown modulate the H3R2 di-methylation at the fibrosis-associated genes' promoter sites? PRMT5 is responsible for the methylation of many histone proteins like histone H2A, H3, H4, and as well as non-histone proteins, why only H3R2 dimethylation examined? Some of them might also be important for the fibroblast activation and regulated by PRMT5, therefore contribute to the observed protective effects of PRMT5 on cardiac fibrosis.

Response: We appreciate the reviewer's comment. Without TGF- β stimulation of cultured cardiac fibroblasts, neither inhibition nor knockdown of PRMT5 suppressed fibrotic gene expression as measured by qPCR analysis (revised Figures 2c, e, and g). In addition, we have checked H3R2 dimethylation with ChIP analysis. Consistent with the

results of qPCR analysis, H3R2 dimethylation was not significantly altered by treatment with the PRMT5 inhibitor (revised Figures 2i and j). We have added these data to the revised manuscript.

As indicated by the reviewer, PRMT5 modulates the histone proteins H2R3, H3R2, H3R8, and H4R3. While the dimethylation of H4R3 represses gene expression, dimethylation of H3R2 is known to activate gene expression. As indicated in Figures 1 and 2, in our study, loss of PRMT5 function suppressed fibrotic gene expression, suggesting that PRMT5 is required to promote fibrotic gene expression. Therefore, we hypothesized that PRMT5-mediated H3R2 dimethylation is important for fibrotic gene transcription. We have examined H4R3 dimethylation, which is a representative PRMT5 modification of histone and is known to repress gene transcription. The results showed that H4R3 dimethylation was not significantly altered by TGF- β stimulation. Huang M et al reported the similar phenomenon of increased dimethylation of H3R2 on LEF1 promoters without increased dimethylation of H4R3. We have added these results to Supplemental Figure 11 and to the Results section as follows:

“We also examined H4R3 dimethylation, which is a representative PRMT5 modification of histone and is known to repress gene transcription¹⁹. The results showed that H4R3 dimethylation was not significantly altered by TGF- β stimulation (Supplementary Figure 11).” (Pages 7-8, Lines 145-148)

1. Huang M, Dong W, Xie R, Wu J, Su Q, Li W, Yao K, Chen Y, Zhou Q, Zhang Q, et al. HSF1 facilitates the multistep process of lymphatic metastasis in bladder cancer via a novel PRMT5-WDR5-dependent transcriptional program. *Cancer Communications*. 2022;42:447-470.

Comment 3

To characterize the activation of fibroblast, CF proliferation should be examined. In addition to differentiation, is PRMT5 required for CF proliferation?

Response: We appreciate the reviewer's comment. In response, we have examined CF proliferation with an MTT assay. The results showed that neither PRMT5 inhibition nor knockdown significantly affected CF viability or proliferation. In addition, treatment with the PRMT5 inhibitor EPZ015666 did not significantly affect the viability and proliferation of adult cardiac fibroblasts. We have added this data to Supplementary Figures 8 and 9 and described them in the Results section.

“As the proliferation of fibroblasts is also characteristic of fibroblast activation, we also

examined the proliferation of cardiac fibroblasts after PRMT5 knockdown and inhibition. The results showed that neither PRMT5 inhibition nor knockdown significantly affected fibroblast viability or proliferation (Supplementary Figure 8). In addition, treatment with the PRMT5 inhibitor EPZ015666 did not significantly affect the viability or proliferation of adult cardiac fibroblasts (Supplementary Figure 9).” (Page 7, lines 132-137)

Comment 4

It is interesting that the cardiac hypertrophy was suppressed in the Postn MCM/Prmt5 flox/flox mice but not in the col1a2 MCM/Prmt5 flox/flox mice. The authors should explain. In addition, does PRMT5 regulate the methylation level or expression of Col1a2?

Response: We appreciate the reviewer's comment. McLellan et al. reported the results of a single-cell RNA-seq of mouse hearts treated with angiotensin II in Circulation (1448-1463, 2020)¹. The analysis showed that cardiac fibroblasts could be divided into nine subpopulations and that gene expression was altered among the groups. Postn expression was detected only in hypertrophy-induced activated cells, and the Postn-positive subpopulation was reported to contribute myofibroblast differentiation. However, collagen expression was detected in quiescent fibroblasts. We speculate that there may be some differences between the *Postn*^{MCM/Prmt5 flox/flox} mice and the *col1a2*^{MCM/Prmt5 flox/flox} mice that we used due to the effect of a fibroblast subpopulation. However, the functional role of each subpopulation of cardiac fibroblasts is not yet fully understood. In addition, PRMT5 inhibition did not significantly affect the expression of Col1a2 in cardiac fibroblasts.

1. McLellan MA, Skelly DA, Dona MSI, Squiers GT, Farrugia GE, Gaynor TL, Cohen CD, Pandey R, Diep H, Vinh A, Rosenthal NA, Pinto AR: High-Resolution Transcriptomic Profiling of the Heart During Chronic Stress Reveals Cellular Drivers of Cardiac Fibrosis and Hypertrophy. *Circulation*, 142, 1448-1463 (2020).

Comment 5

Fig 3c-3d, the interaction between SMAD3 and PRMT5 has been verified by IP-WB in HEK293T at basal level. I wonder whether this direct interaction can still exist in cardiac fibroblasts without TGF β stimulation. If not, why? If so, what is the upstream regulatory mechanism that recruited PRMT5 to SMAD3 binding site under TGF β stimulation. Speculatively, if the interaction is specifically happened under TGF β stimulation, Figure S8, does PRMT5 methylate Smad3 in vivo under TGF β stimulation?

Response: We appreciate the reviewer's comment. As indicated by the reviewer, the interaction between SMAD3 and PRMT5 was detected without TGF- β stimulation in cardiac fibroblasts, suggesting that TGF- β stimulation was not a trigger of the interaction. PRMT5 localized in both the cytosol and the nucleus, indicating that it can interact with Smad3 in both cellular components. In response to TGF- β stimulation, SMAD3 translocates from the cytosol to the nucleus and binds to its target sites on genomic DNA such as Acta2¹. It appears that SMAD3-interacted PRMT5 moves to these target sites and methylates histones that regulate fibrotic gene transcription. We have added this data to Supplementary Figure 12.

“In addition, we examined PRMT5-Smad3 interaction with or without TGF- β stimulation in cardiac fibroblasts. This interaction was detected without TGF- β stimulation, and it did not change in the heart after TAC surgery (Supplementary Figure 13), suggesting that TGF- β stimulation was not a trigger of the interaction.” (Page 8 lines 165-168)

1. Akhurst RJ, Hata A. Targeting the TGF β signalling pathway in disease. *Nat Rev Drug Discov*. 2012;11:790.

Comment 6

Figure S9C, it seems that the phosphorylation of SMAD2/3 bands in TGF β +EPZ015666 upregulated? The statistical data in right should be carefully checked. Thus, it should be caution with the interpretation of the currently proposed model. More much evidence should be provided. e.g. the phosphorylation level of SMAD3 in Prmt5 knockdown mice

with and without TAC surgery.

Response: We appreciate the reviewer's comment. The phosphorylation of the SMAD2/3 bands in TGF β +EPZ015666 was not significantly upregulated in our experiments (p=0.9989, n=5). We have revised the representative images in Supplementary Figure 15c and the data in Supplementary Figure 15.

We have also examined the phosphorylation level of SMAD3 in mouse hearts, and found that there was no difference between WT and PRMT5 KO groups. We have added the data to Supplementary Figure 16.

“Additionally, we examined the phosphorylation level of SMAD3 in mouse heart, and found that there was no difference between the WT and PRMT5 KO groups (Supplementary Figure 16).” (Page 9 lines 186-188)

Comment 7

The authors claims that PRMT5-mediated H3R2 dimethylation is required for myofibroblast differentiation in cardiac fibroblasts. However, limited data directly support this point. The data currently only support that PRMT4 can affect the deposition of H3R2 dimethylation on Colla1 and Acta2, the expression of Colla1 and Acta2 are regulated by intact PRMT5. However, whether the expression change of Colla1 and Acta2 are directly regulated by PRMT5-mediated H3R2 dimethylation is unknown.

Response: We appreciate the reviewer's comment. As suggested by the reviewer, our data only indicate that PRMT5 increased H3R2 dimethylation, and loss of function of PRMT5 suppressed myofibroblast differentiation in cardiac fibroblasts. The data in this study show that (1) PRMT5 was recruited onto the promoter sites of Colla1 and Acta2, (2) H3R2 dimethylation was induced by TGF- β stimulation, and (3) loss of function of PRMT5 decreased the TGF- β -induced expression of Acta2, a molecular marker of myofibroblast differentiation. These data suggest that our hypothesis may be correct, but the evidence is not conclusive. An experimental technique to specifically deplete histone methylation in cells has not yet been established. We have revised the text as follows.

“These results suggest that PRMT5-mediated dimethylation of H3R2 may promote TGF- β -induced fibrotic gene transcription.” (Page 8, lines 148-149)

“As the above results suggested that H3R2 dimethylation was induced by TGF- β stimulation of cardiac fibroblasts, we next considered the role of H3K4 trimethylation in fibrotic gene transcription.” (Page 10, lines 197-199)

Comment 8

PRMT5 mediated H3R2me2s modification may activate transcription via recruitment of WDR5/MLL and the subsequent H3K4me3 modification. It may also function via other pathway. Fig. 4, even the H3K4 trimethylation deposition was affected by PRMT5 inhibitor, we cannot get the conclude that H3K4 trimethylation is mediated by PRMT5-induced H3R2 dimethylation. To support the proposed conclusion in this section, more experiment data should be provided. It would be better to provide data to confirm (1) the H3K4 trimethylation change in fibroblast is directly regulated by H3R2 dimethylation on those examined genes, like Cola1, Acta2. (2) the contribution of TGF β stimulation should be clarified, whether H3K4 trimethylation deposition which is induced by H3R2 dimethylation are only happened in myofibroblast differentiation (or TGF β -stimulation dependent) or not?

Response: We appreciate the reviewer's comment. Our conclusion that PRMT5-mediated H3R2me2s modification may activate transcription via recruitment of WDR5/MLL and the subsequent H3K4me3 modification is based on previous reports, including that by Migliori V et al. (Nat Struct Mol Biol 2012, 19, 136-44)¹. That paper showed that WDR5 recognizes symmetric dimethylation of H3R2 by PRMT5. In addition, other groups have previously shown that H3K4 trimethylation deposition is induced by H3R2 dimethylation in the promoter region under some pathological conditions²⁻⁴.

As described in our response to Comment 7 above, the dimethylation of histones in cardiac fibroblasts cannot be controlled in a specific region due to technical limitations. Because of this, Huang M et al showed that H3K4 trimethylation and H3R2 dimethylation in UM-UC-3 bladder cancer cells was decreased by PRMT5 knockdown in order to demonstrate PRMT5-WDR5-mediated histone modifcaiton⁴. Our data in Figure 4b shows that treatment with a PRMT5 inhibitor reduced TGF- β -induced H3K4 trimethylation in cardiac fibroblasts. We also showed in Supplementary Figure 18 that treatment with a PRMT5 inhibitor did not affect total amount of H3K4 trimethylation, which indicates WDR5/MLL1 methyltransferase activity.

Also, our data showed that TGF- β stimulation increases H3K4 trimethylation (Figures 4a and b) and H3R2 dimethylation (Figures 2e, f and 3g). Previous research by other groups has shown that H3K4 trimethylation deposition induced by H3R2 dimethylation also occurs in other pathologies²⁻⁴ (cell invasion, metabolic syndrome, bladder cancer, etc.).

Considering these previous reports, our data suggest that H3K4 trimethylation is mediated by PRMT5-induced H3R2 dimethylation, but we cannot definitively draw this

conclusion, as pointed out by the reviewer. We have therefore revised the text as follows:
“These findings suggest that PRMT5 regulates TGF- β /Smad3-dependent fibrotic gene transcription, possibly through histone methylation crosstalk, and plays a critical role in cardiac fibrosis and dysfunction.” (Page 2, lines 19-21)

“In addition to PRMT5 recruitment, H3R2 dimethylation at *Colla1* and *Acta2* promoter sites induced by TGF- β stimulation was significantly inhibited by Smad3 knockdown (Figure 3g).” (Page 9, lines 180-182)

“These results suggest that TGF- β -induced H3K4 trimethylation may be mediated by PRMT5-induced H3R2 dimethylation, which is recognized by the WDR5/MLL1 complex, in cardiac fibroblasts.” (Page 10, lines 211-214)

“(3) that both PRMT5-mediated histone arginine methylation and WDR5/MLL1-mediated histone lysine methylation play an important role in fibrotic gene transcription in cardiac fibroblasts” (Page 12, lines 258-260)

“PRMT5-induced arginine methylation and MLL1-induced lysine methylation may then initiate transcription of the α -SMA gene, inducing myofibroblast differentiation.” (Pages 15-16, lines 346-348)

1. Migliori V, Muller J, Phalke S, Low D, Bezzi M, Mok WC, Sahu SK, Gunaratne J, Capasso P, Bassi C, et al. Symmetric dimethylation of H3R2 is a newly identified histone mark that supports euchromatin maintenance. *Nat Struct Mol Biol.* 2012;19:136-144.
2. Chen H, Lorton B, Gupta V, Shechter D. A TGF β -PRMT5-MEP50 axis regulates cancer cell invasion through histone H3 and H4 arginine methylation coupled transcriptional activation and repression. *Oncogene.* 2017;36:373-386.
3. Tsai W-W, Niessen S, Goebel N, Yates JR, Guccione E, Montminy M. PRMT5 modulates the metabolic response to fasting signals. *Proc Natl Acad Sci USA.* 2013;110:8870-8875.
4. Huang M, Dong W, Xie R, Wu J, Su Q, Li W, Yao K, Chen Y, Zhou Q, Zhang Q, et al. HSF1 facilitates the multistep process of lymphatic metastasis in bladder cancer via a novel PRMT5-WDR5-dependent transcriptional program. *Cancer Communications.* 2022;42:447-470.

Minor:

Comment 1

Fig 1d, it seems that *Col1a2* MCM/Prmt5 flox/flox mice decreased fibrosis at Sham group. Is it statistically significant?

Response: We appreciate the reviewer's comment. The result was not statistically significant. There was no significant difference in fibrotic area between the *Prmt5*^{fllox/fllox} mice and the *Colla2*^{MCM}/*Prmt5*^{fllox/fllox} mice at Sham group. The detailed results of statistical analysis are as follows (two-way ANOVA, Tukey's test): Predicted mean diff. = 0.3417, 95%CI = -1.020 to 1.703, P value = 0.8981.

Reviewer #2 (Remarks to the Author):

In this paper entitled "Fibroblast-specific PRMT5 deficiency suppresses pressure overload-induced cardiac fibrosis and left ventricular dysfunction", authors declared that PRMT5 regulates TGF- β /Smad3-dependent fibrotic gene transcription through histone methylation crosstalk and plays a critical role in cardiac fibrosis and dysfunction. However, the data are not elucidating the contribution of fibroblast for the observed phenotypes and several results are hard to interpret with the given information. Specific comments can be found below.

Response: We are grateful to Reviewer 2 for the critical comments and valuable suggestions that have helped us improve our paper. As indicated in the following responses, we have incorporated all of these comments and suggestions in the revised version of our manuscript.

Major concern:

Comment 1

PostnMCM/Prmt5^{flox/flox} and Col1a2MCM /Prmt5^{flox/flox} mice were performed in Figure 1; however, it is required for the baseline level of mouse organs, including the heart, and the fluorescence and expression evidence of fibroblast specific knockout.

Response: We appreciate the reviewer's comment. We have added data on the fibroblast-specific knockout of PRMT5 in *Postn*^{MCM}/*Prmt5*^{flox/flox} mice (Figures 1b and c). The baseline heart weight/body weight ratio did not change, as shown in Figure 1e. We have also shown the specific knockout of *Prmt5* in *Col1a2*^{MCM}/*Prmt5*^{flox/flox} mice in Supplementary Figure 3.

“The PRMT5 floxed mice used in our experiment have previously been reported to reduce PRMT5 protein expression efficiently when crossed with CreERT mice after tamoxifen injection^{17,23}. We examined the deletion of the PRMT5 gene with PCR, and western blotting showed that the expression of PRMT5 in fibroblasts was decreased (Figures 1b, c).” (Page 5 lines 93-97)

“While the baseline heart weight/body weight ratio was not changed by *Prmt5* knockout in fibroblasts, TAC-induced cardiac hypertrophy was suppressed in the *Postn*^{MCM} /*Prmt5*^{flox/flox} mice (Figure 1e, Supplementary Figure 2).” (Page 6 lines 99-102)

Comment 2

Cre mice have cardiotoxicity, so a group of cre mice need to be added as the control.

Response: We appreciate the reviewer's comment. Some reports have shown that MerCreMer expression induces cardiotoxicity, specifically in cardiomyocytes using a Myh6 promoter. Kanisicak O et al. generated Postn gene-targeted mice and reported that MerCreMer expression was not detected in cardiomyocytes using GFP reporter mice¹. Moreover, *Postn*^{MCM} mice have not been reported to show significant changes in cardiac function². These reports suggest that the Postn-induced MerCreMer mice used in our experiments did not induce cardiotoxicity and therefore are unlikely to have affected our experimental results.

1. Kanisicak O, Khalil H, Ivey MJ, Karch J, Maliken BD, Correll RN, Brody MJ, SC JL, Aronow BJ, Tallquist MD, et al. Genetic lineage tracing defines myofibroblast origin and function in the injured heart. *Nat Commun.* 2016;7:12260.
2. Kaur H, Takefuji M, Ngai CY, Carvalho J, Bayer J, Wietelmann A, Poetsch A, Hoelper S, Conway SJ, Mollmann H, et al. Targeted Ablation of Periostin-Expressing Activated Fibroblasts Prevents Adverse Cardiac Remodeling in Mice. *Circ Res.* 2016;118:1906-1917.

Comment 3

Cardiac Echocardiography data display is insufficient, such as left ventricular mass index.

Response: We appreciate the reviewer's comment. We have added the cardiac echocardiography data, including the LV mass index, to Supplementary Tables 1 and 2. We have also revised the method described in our manuscript as follows: "LV mass was calculated as $1.055 [(IVSd + LVIDd + LVPWT)^3 - (LVIDd)^3]$. LV mass index is represented as the ratio of LV mass to body weight." (Page 18 lines 398-400)

Comment 4

The authors only evaluated the level of fibrosis in Figure 1, ignoring myocardial hypertrophy. Whether *Postn*MCM/*Prmt5*flox/flox and *Col1a2*MCM /*Prmt5* flox/flox mice subjected TAC displayed myocardial hypertrophy compared with flox mice ?

Response: We appreciate the reviewer's comment. We showed the data on myocardial hypertrophy (heart weight/body weight and cardiomyocyte surface area) in Supplementary Figures 4 and 5 of the original version. Although *Prmt5* knockout in

Postn-positive fibroblasts inhibited cardiac hypertrophy to a statistically significant degree (figure moved from Supplementary Figure 4 to Figure 1e) compared with control mice, this inhibition may not have been sufficient to improve cardiac function. However, Prmt5 knockout in Colla2-positive fibroblasts did not significantly suppress cardiac hypertrophy compared with control mice. We have made the following changes to the text:

“While the baseline heart weight/body weight ratio was not changed by *Prmt5* knockout in fibroblasts, TAC-induced cardiac hypertrophy was suppressed in the *Postn^{MCM}/Prmt5^{flox/flox}* mice (Figure 1e, Supplementary Figure 2).” (Page 6 lines 99-102)

“Cardiac hypertrophy was not suppressed in the *Colla2^{MCM}/Prmt5^{flox/flox}* mice (Supplementary Figure 5).” (Page 6 lines 106-107)

Comment 5

α -SMA is the marker of myofibroblasts. It is also expressed in mural cells, and more markers are needed as fibroblast marker, refer to the article (Cell Rep. 2018 Jan 16;22(3):600-610. doi: 10.1016/j.celrep.2017.12.072.)

Response: We appreciate the reviewer's comment. As the reviewer indicated, α -SMA is considered a marker of myofibroblasts but is also strongly expressed in vascular smooth muscle cells. α -SMA-expressed stromal cells are considered myofibroblasts, which play an important role in the development of tissue fibrosis. We may not have made ourselves clear in our original paper. While we are also interested in fibroblast function, our main focus is on the development of fibrosis. That is why we used not only α -SMA but also Colla1, a well-known marker of fibrosis.

Comment 6

The role of PRMT5 in phenotypic transformation of fibroblasts was showed in Figure2; actually, the fibrotic phenotype is not only phenotypic transformation, but also includes proliferation and collagen contraction.

Response: We appreciate the reviewer's comment. As suggested by the reviewer, the fibrotic phenotype is not the only phenotypic transformation. In response, we have examined CF proliferation with an MTT assay. The results showed that PRMT5 inhibition and knockdown did not significantly affect CF proliferation. We have added this data to Supplementary Figure 8 and 9.

“As the proliferation of fibroblasts is also characteristic of fibroblast activation, we also

examined the proliferation of cardiac fibroblasts after PRMT5 knockdown and inhibition. The results showed that neither PRMT5 inhibition nor knockdown significantly affected fibroblast viability or proliferation (Supplementary Figure 8). In addition, treatment with the PRMT5 inhibitor EPZ015666 did not significantly affect the viability or proliferation of adult cardiac fibroblasts (Supplementary Figure 9).” (Page 7 lines 132-137)

Comment 7

In the pressure overload-induced cardiac fibrosis model, it is wrong for the author to focus only on the transformation from fibroblasts to myofibroblasts. In the article (Circulation. 2020 Oct 13;142(15):1448-1463. doi: 10.1161/CIRCULATIONAHA.119.045115. Epub 2020 Jul 30.), activated fibroblast emerged after induction of tissue stress to promote fibrosis instead of smooth muscle actin-expressing myofibroblasts, a key profibrotic cell population.

Response: We appreciate the reviewer's comment. We agree that the McLellan study is impressive and that a variety of subpopulations of fibroblasts contribute to the development of cardiac fibrosis. However, alpha smooth muscle actin-expressing myofibroblasts are well-established as a key profibrotic cell population and are widely accepted as the main protagonists driving fibrosis, as described by McLellan et al. themselves. Because of this, we focused on PRMT5 function in myofibroblast differentiation from activated fibroblasts (periostin-expressing fibroblasts).

Comment 8

How did the authors determine smad3 as the target of prmt5? In fact, the fibrosis pathway also includes p38, yap, etc.

Response: We appreciate the reviewer's comment. Smad3 is the primary regulator of TGF- β signaling in TAC-induced cardiac fibrosis, as reported by Khalil H et al. (ref 10 in manuscript). PRMT5 is known as an epigenetic regulator that functions through the modulation of histone and transcription factors. We, therefore, hypothesized that PRMT5 binds to Smad3 and controls TGF- β /Smad3 mediated gene transcription. Moreover, in our study, the use of a PRMT5 inhibitor did not markedly suppress the phosphorylation of p38 in cardiac fibroblasts. We have added this data to Supplementary Figure 17.

"p38 MAPK signaling is another factor known to play an important role in fibroblast activation²¹. However, in our study, the use of a PRMT5 inhibitor did not markedly suppress the phosphorylation of p38 in cardiac fibroblasts (Supplementary Figure

17).“ (Page 9 lines 188-191)

1. Khalil H, Kanisicak O, Prasad V, Correll RN, Fu X, Schips T, Vagnozzi RJ, Liu R, Huynh T, Lee SJ, et al. Fibroblast-specific TGF-beta-Smad2/3 signaling underlies cardiac fibrosis. *J Clin Invest.* 2017;127:3770-3783.
2. Molkenin JD, Bugg D, Ghearing N, Dorn LE, Kim P, Sargent MA, Gunaje J, Otsu K, Davis J. Fibroblast-Specific Genetic Manipulation of p38 Mitogen-Activated Protein Kinase In Vivo Reveals Its Central Regulatory Role in Fibrosis. *Circulation.* 2017;136:549-561.

Comment 9

H3R2 symmetric dimethylation has been reported to induce H3K4 trimethylation; The trimethylation induced fibrosis gene transcription was inhibited by a PRMT5 inhibitor and WDR5/MLL1 inhibitor. These results cannot prove the regulation of PRMT5/WDR5/MLL1 methylase complex. The author first needs to prove that prmt5 regulates WDR5/MLL1 methylase comple, and demethylase also needs to be excluded.

Response: We appreciate the reviewer's comment. We have added to Supplementary Figure 18 data showing that the PRMT5 inhibitor did not alter the total amount of H3K4 trimethylation in cultured cardiac fibroblasts. These data suggest that PRMT5 inhibition may not suppress MLL1 methyltransferase activity. We have made the following changes to the text:

“To determine the effect of PRMT5 on MLL1 methyltransferase activity, we examined total H3K4 trimethylation. The result showed that PRMT5 inhibition did not significantly change the total amount of H3K4 trimethylation in cardiac fibroblasts (Supplementary Figure 18).” (Page 10, lines 208-211)

Our conclusion that PRMT5-mediated H3R2me2s modification may activate transcription via recruitment of WDR5/MLL and subsequent H3K4me3 modification is based on previous reports, including Migliori V et al. (Nat Struct Mol Biol 2012, 19, 136-44). This paper showed that WDR5 recognizes symmetric dimethylation of H3R2 by PRMT5. As pointed out by the reviewer, the description "PRMT5/WDR5/MLL1 methyltransferase complex" is not appropriate. We have revised the text as follows: “These results suggest that the WDR5/MLL1 methyltransferase complex is required for myofibroblast differentiation in cardiac fibroblasts.” (Page 10, lines 220-221)

1. Migliori V, Muller J, Phalke S, Low D, Bezzi M, Mok WC, Sahu SK, Gunaratne J,

- Capasso P, Bassi C, et al. Symmetric dimethylation of H3R2 is a newly identified histone mark that supports euchromatin maintenance. *Nat Struct Mol Biol.* 2012;19:136-144.
2. Chen H, Lorton B, Gupta V, Shechter D. A TGF β -PRMT5-MEP50 axis regulates cancer cell invasion through histone H3 and H4 arginine methylation coupled transcriptional activation and repression. *Oncogene.* 2017;36:373-386.
 3. Tsai W-W, Niessen S, Goebel N, Yates JR, Guccione E, Montminy M. PRMT5 modulates the metabolic response to fasting signals. *Proc Natl Acad Sci USA.* 2013;110:8870-8875.
 4. Huang M, Dong W, Xie R, Wu J, Su Q, Li W, Yao K, Chen Y, Zhou Q, Zhang Q, et al. HSF1 facilitates the multistep process of lymphatic metastasis in bladder cancer via a novel PRMT5-WDR5-dependent transcriptional program. *Cancer Communications.* 2022;42:447-470.

Comment 10

EPZ015666 suppresses pressure overload-induced cardiac fibrosis and dysfunction. EPZ is a prmt5 inhibitor, acting on several kind of cells including fibroblasts. The therapeutic strategy of fibroblast specific targeting PRMT5 needs to be applied.

Response: We appreciate the reviewer's comment. As indicated by the reviewer, EPZ015666 exhibited its action on several kinds of cells in the body. However, a pharmacological therapeutic strategy based on fibroblast-specific targeting of PRMT5 has not yet been developed, which is a limitation of fibroblast-targeted therapy. Additionally, we showed our data on the fibroblast-specific knockout of PRMT5 in mice in Figure 1. These data suggest that PRMT5 plays a role in the development of cardiac fibrosis in fibroblasts.

Reviewer #3 (Remarks to the Author):

In this manuscript the authors nicely demonstrated deletion of Prmt5 in cardiac fibroblasts reduced cardiac fibrosis after TAC-induced hypertrophic remodeling. Furthermore, they have showed that PRMT5 interacted with Smd3 and promoted aSMA expression in response to Tgfb using fibroblast cell culture model. Finally, they showed that pharmacological inhibition of PRMT5 has similar cardiac phenotype as Prmt5 KO.

Response: We are grateful to Reviewer 3 for the critical comments and useful suggestions that have helped us improve our paper. As indicated in the following responses, we have incorporated all of these comments and suggestions in the revised version of our manuscript.

Major concerns:

Comment 1

Hypertrophic remodeling is a hallmark of TAC injury. The authors showed that fibroblast-deletion of Prmt5 reduced TAC-induced cardiac fibrosis. How about post-TAC hypertrophic remodeling? e.g, HW/BW, cardiomyocyte size, hypertrophic gene expression (ANF, BNF, a/bMHC), etc. ? Are they reduced by Prmt5-deletion as well?

Response: We appreciate the reviewer's comment. In the original version, we showed the relevant data in Supplementary Figures 4 and 5. These data are shown in Figure 1e and Supplementary Figures 2 and 5 in the revised manuscript. While hypertrophy was not suppressed in the *Col1a2*-PRMT5-KO mice, it was slightly inhibited in the Postn-PRMT5-KO mice. Although Prmt5 knockout in Postn-positive fibroblasts decreased cardiac hypertrophy to a statistically significant degree, that decrease does not appear to have been sufficient to improve cardiac function. We have made the following changes to the text:

“While the baseline heart weight/body weight ratio was not changed by *Prmt5* knockout in fibroblasts, TAC-induced cardiac hypertrophy was suppressed in the *Postn^{MCM}/Prmt5^{lox/lox}* mice (Figure 1e, Supplementary Figure 2).” (Page 6 lines 99-102)

“Cardiac hypertrophy was not suppressed in the *Col1a2^{MCM}/Prmt5^{lox/lox}* mice (Supplementary Figure 5).” (Page 6 lines 106-107)

Comment 2

Like the authors pointed out in the discussion, PRMT5's function in cardiomyocytes is opposite from that of fibroblast; deletion of Prmt5 in cardiomyocytes caused dilated

cardiomyopathy. What about pharmacological inhibition of PRMT5 using PRMT5 inhibitor EPZ015666? Since EPZ015666 is administered systemically, it should inhibit PRMT5 in cardiomyocytes as well? In figure 5 experiments, do you see changes in OGA expression and cardiac protein O-GlcNacylation caused by EPZ-015666 treatment?

Response: We appreciate the reviewer's comment. In our experiment, treatment with a PRMT5 inhibitor for eight weeks did not result in cardiotoxicity in mice (Figure 5). Many reports have shown that various PRMT5 inhibitors have an anti-cancer effect in *in vivo* experiments, but none have reported severe side effects¹⁻³. Additionally, clinical trials have not reported that PRMT5 inhibitors have severe adverse effects^{4,5}.

We have examined the effect of a PRMT5 inhibitor on cultured cardiomyocytes. EPZ015666 suppressed myocyte hypertrophy induced by phenylephrine. We have also checked cardiomyocyte viability and changes in OGA expression. EPZ015666 did not decrease the cell viability or alter the OGA expression in cardiomyocytes. We have added this data to Supplementary Figures 20 and 21:

“A previous study reported that conditional knockout of PRMT5 in cardiomyocytes caused dilated cardiomyopathy through the dysregulation of O-GlcNAcase (OGA)³⁵. We therefore examined the cell viability of cultured cardiomyocytes from neonatal rats after EPZ015666 treatment. The results did not show a significant decrease in cell viability or OGA expression (Supplementary Figures 20 and 21).” (Pages 11 lines 240-245)

The difference between PRMT5 knockout in cardiomyocytes and treatment with a PRMT5 inhibitor should be investigated in further studies on PRMT5-targeted pharmacotherapy. In a previous study by Li Z et al⁶, the phenotype of dilated cardiomyopathy was shown in seven-month-old mice. We speculate that the duration of the treatment with the PRMT5 inhibitor induced the difference in the phenotype.

1. Strobl CD, Schaffer S, Haug T, Völkl S, Peter K, Singer K, Böttcher M, Mougiakakos D, Mackensen A, Aigner M. Selective PRMT5 Inhibitors Suppress Human CD8(+) T Cells by Upregulation of p53 and Impairment of the AKT Pathway Similar to the Tumor Metabolite MTA. *Molecular cancer therapeutics*. 2020;19:409-419.
2. Fedoriw A, Rajapurkar SR, O'Brien S, Gerhart SV, Mitchell LH, Adams ND, Rioux N, Lingaraj T, Ribich SA, Pappalardi MB, et al. Anti-tumor Activity of the Type I PRMT Inhibitor, GSK3368715, Synergizes with PRMT5 Inhibition through MTAP Loss. *Cancer Cell*. 2019;36:100-114.e125.
3. Yin S, Liu L, Brobbey C, Palanisamy V, Ball LE, Olsen SK, Ostrowski MC, Gan W.

- PRMT5-mediated arginine methylation activates AKT kinase to govern tumorigenesis. *Nat Commun.* 2021;12:3444
4. Ahnert JR, Perez CA, Wong KM, Maitland ML, Tsai F, Berlin J, Liao KH, Wang I-M, Markovtsova L, Jacobs IA, et al. PF-06939999, a potent and selective PRMT5 inhibitor, in patients with advanced or metastatic solid tumors: A phase 1 dose escalation study. *Journal of Clinical Oncology.* 2021;39:3019-3019.
 5. Siu LL, Rasco DW, Vinay SP, Romano PM, Menis J, Opdam FL, Heinhuis KM, Egger JL, Gorman SA, Parasrampur R, et al. 438O - METEOR-1: A phase I study of GSK3326595, a first-in-class protein arginine methyltransferase 5 (PRMT5) inhibitor, in advanced solid tumours. *Annals of Oncology.* 2019;30:v159.
 6. Li Z, Xu J, Song Y, Xin C, Liu L, Hou N, Teng Y, Cheng X, Wang T, Yu Z, et al. PRMT5 Prevents Dilated Cardiomyopathy via Suppression of Protein O-GlcNAcylation. *Circ Res.* 2021;129:857-871.

Minor concerns.

Comment 1

Figure 1g, aSMC staining in sham hearts appears to be from smooth muscle cells around vessel rather than from myofibroblasts. It may be difficult to identify aSMA-expressing myofibroblasts under baseline condition since myofibroblasts are not activated. This may need to be clarified.

Response: We appreciate the reviewer's comment. As indicated by the reviewer, α -SMA was expressed in the smooth muscle cells in the heart and a large amount of α -SMA expression was observed in blood vessels. We can easily detect blood vessels and eliminate them from the analysis. There was a very small number of α -SMA positive cells in the stroma of the heart under baseline conditions (without TAC surgery). In order to clarify the situation, we have added the following sentence to the Method section: " α -SMA positive cells around blood vessels were eliminated from the analysis." (Page 19 lines 441-442)

Comment 2

The mechanistic studies in figure 2-4 were done with primary cardiac fibroblasts. For Nat communication, I think to confirm some of the results in the hearts may be necessary.

Response: We appreciate the reviewer's comment. In response, we have performed the additional experiments and added data to Supplementary Figures 13B and 16. The data

have shown that the interaction between PRMT5 and Smad3 was detected in the hearts with or without TAC surgery (Supplementary Figure 13B) and that the phosphorylation of Smad3 was not altered by Prmt5 knockout in fibroblasts (Supplementary Figure 16). While the ideal method would be to isolate the cardiac fibroblasts *in vivo* and directly analyze the mechanism of PRMT5, specifically isolating PRMT5-KO cardiac fibroblasts requires Cre-recognized reporter mice such as GFP mice.

Reviewer #4 (Remarks to the Author):

The manuscript addresses the interesting and highly relevant therapeutic concept of preventing cardiac remodeling and heart failure by preventing cardiac hypertrophy and fibrosis. The authors demonstrate that a fibroblast specific deletion of PRMT5 protects from adverse left ventricular remodeling after pressure overload induced by TAC in mice. The mechanism is claimed to be that PRMT5 forms a complex with SMAD3 after TGF beta stimulation, which increases symmetric dimethylation of target gene promoters. This gets in turn recognized by WDR5/MLL1 which regulates TGF-beta induced fibrotic gene regulation. Finally, they switch back to an in vivo model and show a beneficial effect of PRMT5 inhibition by EPZ015666 after TAC.

While this is an important topic the authors need to place more emphasis on clearly demonstrating that their main claim, which is the induction of fibrosis and hypertrophy by PRMT5, is true. This would require substantial additional experimental evidence in order to underscore the author's hypothesis.

There are several major points which need to be addresses:

Response: We are grateful to Reviewer 4 for the critical comments and useful suggestions that have helped us improve our paper. As indicated in the responses that follow, we have incorporated all of these comments and suggestions in the revised version of our manuscript.

Comment 1

The authors have to show that the PRMT5 knockout is working! If this is the initial description and use of this floxed mouse line, they need to show the reduction in PRMT5 protein and that no truncated protein is expressed that could be partially functional. If the line has been published previously, the relevant publication showing knockout efficiency must be cited.

Response: We appreciate the reviewer's comment. The PRMT5 floxed mice used in our experiment have previously been reported to reduce PRMT5 protein efficiently when crossed with CreERT mice after tamoxifen injection^{1,2}. There are no reports of truncated PRMT5 protein being detected. In response to the reviewer's comment, we examined the deletion of the PRMT5 gene with PCR, and western blotting showed that the expression of Prmt5 in fibroblasts was decreased. These data indicate that PRMT5 knockout is working as described in the previous reports. We have added the new data to

Supplementary Figure 1 and the references below to our revised manuscript.

“The PRMT5 floxed mice used in our experiment have previously been reported to reduce PRMT5 protein efficiently when crossed with CreERT mice after tamoxifen injection. We examined the deletion of the PRMT5 gene with PCR, and western blotting showed that the expression of PRMT5 in fibroblasts was decreased (Figures 1b, c).” (Page 5 lines 93-97)

1. Zhang T, Günther S, Looso M, Künne C, Krüger M, Kim J, Zhou Y, Braun T. Prmt5 is a regulator of muscle stem cell expansion in adult mice. *Nat Commun.* 2015;6:7140.
2. Hamard P-J, Santiago GE, Liu F, Karl DL, Martinez C, Man N, Mookhtiar AK, Duffort S, Greenblatt S, Verdun RE, et al. PRMT5 Regulates DNA Repair by Controlling the Alternative Splicing of Histone-Modifying Enzymes. *Cell Reports.* 2018;24:2643-2657

Comment 2

How efficient is the tamoxifen-induced knockout with the POSTN and the Col1a2 MerCreMer lines in the authors' hands after TAC surgery? What's the percentage of myofibroblasts in which the PRMT5 gene was deleted? This is important information to assess the role of PRMT5 in preventing heart remodeling after pressure overload. Also, did the controls receive tamoxifen as well (either in their chow or by injection)?

Response: We appreciate the reviewer's comment. The data in our original submission showed that the knockout efficiency of PRMT5 in total cardiac fibroblasts was approximately 70% in *Col1a2^{MCM}* mice (Supplementary Figure 3d), and we have added new data showing an efficiency of approximately 50% in *Postn^{MCM}* mice (Figure 1c). As suggested by the reviewer, the extent to which PRMT5 was deleted in myofibroblasts is important consideration, but our experiments using knockout mice do not allow us to make that measurement. All of the mice in our experiments were treated with tamoxifen. We have added this information to Method section in our revised manuscript.

"All of the mice in our experiments were treated with tamoxifen, according to the same protocol." (Page 17 lines 379-380)

Comment 3

A major concern is the huge variability of the PRMT5fl/fl control group after TAC! This is true for all experiments, and this reviewer wonders if the surgery was not performed

consistently. What was the survival rate after TAC with a 27G needle? The degree of interstitial fibrosis is very low for a TAC with a 27G needle. Why is this so? Also FS should be less than 30% after 8 weeks. It would be helpful to correlate individual data points (colour coded), so that it is possible to see the values of the outliers in Figs 1 and 5! Is the large variability in the data maybe due to a difference between the sexes? The authors need to indicate whether they used exclusively male or female mice. In Fig. 5 it is mentioned that 8-10 weeks old male C57BL/6j mice were used. What about Fig. 1 where the largest variability takes place?

Response: We appreciate the reviewer's comment. The survival rate after TAC with a 27G needle was 8/10 (PRMT5^{fl/fl} control group) and 6/8 (Postn^{MCM};PRMT5^{fl/fl} control group). This rate may be somewhat low compared to the outcomes of professional surgeons, but we do not think that it affects our conclusion. In response to the reviewer's comment, we have changed the colors of the data points of the control group in Figures 1 and 5 to distinguish each individual mouse; one color represents one mouse. It appears that the wide variation in the data is correlated with individual mice. Additionally, we have statistically analyzed the outliers using ROUT method¹ with GraphPad Prism software. Although a few points were identified as outliers, temporarily excluding them did not change the statistical significance of the results.

The degree of fibrosis may vary among reports due to differing methods of fibrosis evaluation, including image analysis and staining methods. A methodological report by Wang et al. has shown that the fibrosis area in the heart at eight weeks after TAC surgery is approximately 2-3% (figure below)². We evaluated only interstitial fibrosis by removing the perivascular fibrosis area. We have added this information to the Methods in the revised manuscript:

“Interstitial fibrosis is calculated by removing the perivascular fibrosis area and expressed as a percentage of the total tissue section area.” (Page 18 line 417-418)

We used male C57BL/6 mice in all experiments, so the variability is not due to a difference in sex. We have added this information to the Methods in the revised manuscript:

“All of the mice in the experiments were male.” (Page 17 line 376)

Figure 9B. Quantification of the fibrotic area of Masson trichrome stained heart tissues after TAC surgery.

1. Motulsky HJ, Brown RE. Detecting outliers when fitting data with nonlinear regression – a new method based on robust nonlinear regression and the false discovery rate. *BMC Bioinformatics*. 2006;7:123.
2. Wang X, Zhu X, Shi L, Wang J, Xu Q, Yu B, Qu A. A time-series minimally invasive transverse aortic constriction mouse model for pressure overload-induced cardiac remodeling and heart failure. *Frontiers in Cardiovascular Medicine*. 2023;10.

Comment 4

for the in-vitro experiments primary cardiac fibroblasts were taken from 1 to 3 days old rats. Why did the authors change the model system from mouse to rat? Why were fetal cardiac fibroblasts used? Are they a good model for adult cardiac fibrosis? Postnatal fibroblasts are different from adult fibros in terms of expression, proliferative capacity etc. To draw appropriate conclusions, the authors need to verify the data obtained with neonatal rat fibroblasts with adult cardiac fibroblasts. This is an important concern because all mechanistic conclusions were drawn from in vitro experiments.

Response: We appreciate the reviewer’s comment. We chose primary cardiac fibroblasts taken from 1- to 3-day-old rats because a sufficiently large number of cells can be obtained from fetal rats, but not from adult mice. We needed a large number of fibroblasts, without the long-term culturing process, study the differentiation fibroblasts into myofibroblasts.

In response to the reviewer’s comment, we have verified the data using primary cultured human adult cardiac fibroblasts. We have added the results to Figures 2a-d. Pharmacological inhibition and knockdown of PRMT5 significantly suppressed TGF- β -induced α -SMA expression, consistent with our results using neonatal rat cardiac fibroblasts. Cell proliferation of adult cardiac fibroblasts induced by treatment with PDGF

and FGF-2 was not significantly inhibited by treatment with EPZ015666. These data suggest PRMT5 is involved in both neonatal and adult cardiac fibroblasts. These results are shown in Supplementary Figure 9, along with the following change to the text:

“In addition, treatment with the PRMT5 inhibitor EPZ015666 did not significantly affect the viability or proliferation of adult cardiac fibroblasts (Supplementary Figure 9).” (Page 7 lines 135-137)

Comment 5

The number of α SMA-positive fibroblasts was increased after TAC surgery in the controls but to a lesser degree in the Postn-specific Prmt5-knockouts. What is the cause of this phenomenon? Is fibroblast proliferation affected or is apoptosis increased?

the authors need to stain for markers of proliferation and apoptosis and to quantify their results. It is possible that PRMT5 induces fibroblast proliferation, as has been shown in retinoblastoma, and that its main action is via cell cycle genes rather than genes involved in fibrosis. This is very likely because fewer myofibroblasts mean less fibrosis.

Response: We appreciate the reviewer’s comment. As α -SMA is a marker of myofibroblasts, we think that PRMT5 positively regulates myofibroblast differentiation from quiescent cardiac fibroblasts (α -SMA-negative fibroblasts). Our results *in vivo* and *in vitro* showed that loss of PRMT5 function suppressed the expression of α -SMA in cardiac fibroblasts.

In response to the reviewer’s comment, we have examined fibroblast proliferation in mouse heart using Ki-67 and vimentin immunofluorescent staining. Although we could detect Ki-67 positive cells in tumor tissue (considered as a positive control tissue), we did not detect any proliferative fibroblasts (Ki-67- positive/vimentin-positive cells) in the hearts of the mice at eight weeks after TAC surgery, as shown in the figure below. Moreover, we have also examined fibroblast viability after treatment with a PRMT5 inhibitor *in vitro*, and did not detect a decrease in the number of viable fibroblasts (Supplementary Figures 8 and 9). These results indicate that PRMT5 did not affect the proliferation or the cell death of cardiac fibroblasts.

Comment 6

Is there an effect on capillary density in the PRMT5 knockouts? It has been shown that a better vascularization can prevent a maladaptive response.

Response: We appreciate the reviewer’s comment. As indicated by the reviewer, vascularization is a factor known to affect myocyte contraction. In response, we have examined the capillary density in the hearts of PRMT5 knockout mice after TAC surgery. The results show that PRMT5 knockout in fibroblasts did not significantly change capillary density. We have added this data to Supplementary Figure 6:

“We also examined cardiac vascularization, a factor known to affect myocyte contraction. The results showed that PRMT5 knockout in fibroblasts did not significantly change capillary density (Supplementary Figure 6).” (Page 6 lines 109-112)

Comment 7

There is an issue with some of the statistics. In Fig. 2b,d; 4d,e,f and Suppl. Fig.9B the data had been normalized to control samples, which had been given arbitrary values of 1 with zero variance. This data is unsuitable for a one-way ANOVA. The authors need to normalize correctly.

Response: We appreciate the reviewer’s comment. In response, we have normalized the data as well as the other figures. These correctly normalized data were statistically analyzed by one-way ANOVA. We have revised the figures (2b,d; 4d,e,f and Suppl. Fig.9b) in our manuscript.

Minor points:

Comment 1

Complete immunoblot images are missing. Band size markers are missing for all immunoblots.

Response: We appreciate the reviewer's comment. In response, we have submitted the source data, including immunoblot figures.

Comment 2

Fig. 4e Labeling of the blots does not match labeling of the quantification.

Response: We appreciate the reviewer's comment. We have revised Figure 4e as follows:

Comment 3

EMMI: 07883 should be EMMA.

Response: We appreciate the reviewer's comment. We have revised the Method section of the manuscript as follows: EMMA ID: 07883. (Page 17, line 372)

Comment 4

I found a preprint of the manuscript deposited and publicly available in "Research Square", so that a double blind review was not possible.

Response: We appreciate the reviewer's comment. We understand that the review process is not double blinded.

REVIEWER COMMENTS

Reviewer #1 (Remarks to the Author):

The authors have addressed most of my questions. However, there are still some questions in relation to the revised manuscript.

1. Comments to question 1 in author response: As PRMT5 expression does not change in either cardiomyocytes or cardiac fibroblasts under heart failure conditions, it is strange to study the role of PRMT5 in cardiac fibrosis. The authors need to add more clarifications.

[EDITOR: please address.]

2. Comments to question 5 in author response: As TGF- β stimulation was not a trigger of the PRMT5-Smad3 interaction, what is the upstream regulatory mechanism that recruited PRMT5 to SMAD3 binding site under TGF β stimulation?

[EDITOR: Please address, perhaps by discussing rather than including additional data.]

3. Heart weight / BW was provided in Figure 1, while HW/TL in figure 5. Why? Does EPZ015666 treatment alter body weight in mice?

[EDITOR: We hope you can address this with existing data you have obtained.]

4. Regarding to echocardiography parameters, heart rate data should be provided.

[EDITOR: We hope you can address this with existing data you have obtained.]

Reviewer #2 (Remarks to the Author)

The authors have provided all my concerns, and I have no further question.

Reviewer #3 (Remarks to the Author):

Authors have sufficiently addressed my concerns.

Reviewer #4 (Remarks to the Author):

The manuscript has improved significantly, but there is still one point that I don't think has been fully addressed:

Quantification of Ki67-positive cells was performed only on sections from hearts 8 weeks after TAC. However, a pro-proliferative effect would be expected much earlier. Thus, earlier time points should also be analyzed. The same applies to apoptosis.

[EDITOR: Do you have any mouse tissue from earlier timepoints? If not, it might not be worth running additional mouse experiments to address this due to 3Rs concerns.]

The measurement of apoptosis has been only indirect and in vitro. Please provide TUNEL or cleaved Caspase 3 staining on sections from TAC hearts to exclude a change in apoptosis rate. Otherwise, it cannot be concluded that Prmt5 affects cardiac fibroblast development and neither proliferation nor programmed cell death.

[EDITOR: We hope you have sufficient sections to be able to do this as well.]

Regarding the supplementary data, it would be beneficial if the blots were provided in full size and the size markers were provided with lines. The boxes marking the quantified lanes are not set very accurately in all cases. Please correct this.

Responses to Reviewers

Ref.: Manuscript ID NCOMMS-22-45978-R1

Fibroblast-specific PRMT5 deficiency suppresses pressure overload-induced cardiac fibrosis and left ventricular dysfunction

Reviewer 1

Reviewer 1: General comment

The authors have addressed most of my questions. However, there are still some questions in relation to the revised manuscript.

Response: We are grateful to Reviewer 1 for the critical comments and useful suggestions that have helped us improve our paper. As indicated in the following responses, we have incorporated all of these comments and suggestions in the revised version of our manuscript.

Comment 1

Comments to question 1 in author response: As PRMT5 expression does not change in either cardiomyocytes or cardiac fibroblasts under heart failure conditions, it is strange to study the role of PRMT5 in cardiac fibrosis. The authors need to add more clarifications. [EDITOR: please address.]

Response: We appreciate the reviewer's comment. Previous studies have shown that PRMT5 expression is increased under some conditions, such as cancer; however, a change in the amount of PRMT5 expression is not required for PRMT5 function to regulate gene transcription. For example, Wei H, et al have reported that PRMT5 regulates NF- κ B activity without PRMT5 expression being altered¹.

We think that PRMT5 works in association with SMADs. PRMT5 is a type II enzyme that catalyzes arginine methylation and epigenetically regulates the transcription of various genes. It has been reported that epigenetic regulation is essential for myofibroblast differentiation in cardiac fibrosis². The TGF- β /SMAD pathway is known to be a key regulator of myofibroblast differentiation. This pathway also plays a role in epithelial-to-mesenchymal transition (EMT) in cancer cells. A previous report showed that PRMT5 regulates EMT in cancer cells as an epigenetic modifier of TGF- β signaling³. Based on this background, we hypothesized that PRMT5 plays a role in TGF- β -stimulated cardiac fibroblasts and chose to study the function of PRMT5 in cardiac fibrosis. We have revised

the Introduction section to explain our hypothesis more clearly as follows:

"Chen H. et al. reported that the PRMT5/MEP50 methylome complex regulates TGF- β -mediated epithelial-to-mesenchymal transition through histone arginine methylation²⁰. In addition, the TGF- β /Smad3 signal pathway has been shown to be essential for myofibroblast differentiation in tissue fibrosis⁹. However, the role of PRMT5 in cardiac fibrosis during the development of heart failure is still unknown." (Pages 4-5, Lines 24-1)

1. Wei H, Wang B, Miyagi M, She Y, Gopalan B, Huang D-B, Ghosh G, Stark GR, Lu T. PRMT5 dimethylates R30 of the p65 subunit to activate NF- κ B. *Proc Natl Acad Sci USA*. 2013;110:13516-13521.
2. Felisbino MB, McKinsey TA. Epigenetics in Cardiac Fibrosis: Emphasis on Inflammation and Fibroblast Activation. *JACC Basic Transl Sci*. 2018;3:704-715.
3. Chen H, Lorton B, Gupta V, Shechter D. A TGF β -PRMT5-MEP50 axis regulates cancer cell invasion through histone H3 and H4 arginine methylation coupled transcriptional activation and repression. *Oncogene*. 2017;36:373-386.

Comment 2

Comments to question 5 in author response: As TGF- β stimulation was not a trigger of the PRMT5-Smad3 interaction, what is the upstream regulatory mechanism that recruited PRMT5 to SMAD3 binding site under TGF β stimulation?

[EDITOR: Please address, perhaps by discussing rather than including additional data.]

Response: We appreciate the reviewer's comment. Our data showed that TGF- β stimulation did not alter PRMT5-Smad3 interaction in cardiac fibroblasts. TGF- β stimulation is a trigger that translocates Smad3 from the cytoplasm to the nucleus and recruits Smad3 to the DNA¹. We think that PRMT5 is recruited to Smad-binding sites in a Smad3-dependent manner after TGF- β stimulation.

PRMT5 is localized to both the cytoplasm and nucleus. We hypothesized that either (i) PRMT5 interacts with Smad3 in the cytoplasm, and the complex moves to the DNA after TGF- β stimulation, or (ii) Smad3 translocates to the nucleus after TGF- β stimulation and interacts with PRMT5 in the nucleus. However, it remains unclear which possibility is correct.

1. Meng X-m, Nikolic-Paterson DJ, Lan HY. TGF-β: the master regulator of fibrosis. Nat Rev Nephrol. 2016;12:325.

Comment 3

Heart weight / BW was provided in Figure 1, while HW/TL in Figure 5. Why? Does EPZ015666 treatment alter body weight in mice?

[EDITOR: We hope you can address this with existing data you have obtained.]

Response: We appreciate the reviewer's comment. Tibia length is used to normalize body size as well as body weight, but it was not affected by surgery or drug treatment. We have changed the data in Figure 1 to HW/TL. These data indicate the same conclusion as the original version of our manuscript.

There was no significant change in body weight after treatment with EPZ015666. We have revised Figure 1e and added the body weight data to Supplementary Figure 20 as follows.

“There was no significant change in body weight after treatment with EPZ015666 (Supplementary Figure 20).” (Page 11, Lines 10-11)

Comment 4

Regarding to echocardiography parameters, heart rate data should be provided.

[EDITOR: We hope you can address this with existing data you have obtained.]

Response: We appreciate the reviewer's comment, and we agree that the heart rate data

are important. We have added the heart rate data to Supplementary Tables 1 and 2 in the revised manuscript. There were no significant differences between the PRMT5 floxed and PRMT5-KO groups after TAC surgery.

Reviewer #2 (Remarks to the Author):

The authors have provided all my concerns, and I have no further question.

Response: We are grateful to Reviewer 2 for the critical comments and valuable suggestions that have helped us improve our manuscript.

Reviewer #3 (Remarks to the Author):

Authors have sufficiently addressed my concerns.

Response: We are grateful to Reviewer 3 for the critical comments and useful suggestions that have helped us improve our manuscript.

Reviewer #4 (Remarks to the Author):

The manuscript has improved significantly, but there is still one point that I don't think has been fully addressed:

Response: We are grateful to Reviewer 4 for the critical comments and useful suggestions that have helped us improve our paper. As indicated in the responses that follow, we have incorporated all of these comments and suggestions in the revised version of our manuscript.

Comment 1

Quantification of Ki67-positive cells was performed only on sections from hearts 8 weeks after TAC. However, a pro-proliferative effect would be expected much earlier. Thus, earlier time points should also be analyzed. The same applies to apoptosis.

[EDITOR: Do you have any mouse tissue from earlier timepoints? If not, it might not be worth running additional mouse experiments to address this due to 3Rs concerns.]

Response: We appreciate the reviewer's comment. As pointed out by the reviewer, fibroblast proliferation might occur at earlier time points (one week after TAC surgery)¹. However, we do not have any mouse heart tissue at that time point. As pointed out by the editor, we could not perform additional mouse experiments to address this due to 3R concerns. We have revised our manuscript as follows:

“The results showed that neither PRMT5 inhibition nor knockdown significantly affected fibroblast viability or proliferation *in vitro* (Supplementary Figure 8). In addition, treatment with the PRMT5 inhibitor EPZ015666 did not significantly affect the viability or proliferation of adult cardiac fibroblasts *in vitro* (Supplementary Figure 9).” (Page 7, lines 11-15).

“As cardiac fibrosis is regulated by various factors, including fibroblast proliferation, further mechanistic studies are needed to analyze the fibrosis-related functions of PRMT5.” (Page 13, lines 17-19).

1. Moore-Morris T, Guimaraes-Camboa N, Banerjee I, Zambon AC, Kisseleva T, Velayoudon A, Stallcup WB, Gu Y, Dalton ND, Cedenilla M, et al. Resident fibroblast lineages mediate pressure overload-induced cardiac fibrosis. *J Clin Invest.* 2014;124:2921-2934.

Comment 2

The measurement of apoptosis has been only indirect and in vitro. Please provide TUNEL or cleaved Caspase 3 staining on sections from TAC hearts to exclude a change in apoptosis rate. Otherwise, it cannot be concluded that Prmt5 affects cardiac fibroblast development and neither proliferation nor programmed cell death.

[EDITOR: We hope you have sufficient sections to be able to do this as well.]

Response: We appreciate the reviewer's comment. In response, we have examined fibroblast death in mouse heart using TUNEL and vimentin immunofluorescence staining. We did not detect any apoptotic fibroblasts (TUNEL-positive/vimentin-positive cells) in the hearts of the mice at 8 weeks after TAC surgery, as shown in the figure below. This result suggests that PRMT5 did not affect the proliferation or the cell death of cardiac fibroblasts at 8 weeks after TAC surgery. However, as we did not examine the proliferation and apoptosis of cardiac fibroblasts at earlier time points, this is not a conclusive finding, as pointed out by the reviewer. We have revised our manuscript as follows:

“As cardiac fibrosis is regulated by various factors, including fibroblast proliferation, further mechanistic studies are needed to analyze the fibrosis-related functions of PRMT5.” (Page 13, lines 17-19).

Comment 3

Regarding the supplementary data, it would be beneficial if the blots were provided in full size and the size markers were provided with lines. The boxes marking the quantified lanes are not set very accurately in all cases. Please correct this.

[EDITOR: Do make these changes].

Response: We appreciate the reviewer's comment. We have revised the blot figures in the supplementary data to make them full size with lines showing size markers.